# Natural Polymeric Nanobiocomposites for Anti-Cancer Drug Delivery Therapeutics: A Recent Update

**DOI:** 10.3390/pharmaceutics15082064

**Published:** 2023-07-31

**Authors:** Arijit Mondal, Amit Kumar Nayak, Prithviraj Chakraborty, Sabyasachi Banerjee, Bankim Chandra Nandy

**Affiliations:** 1Department of Pharmaceutical Chemistry, M.R. College of Pharmaceutical Sciences and Research, Balisha 743 234, India; 2Department of Pharmaceutics, School of Pharmaceutical Sciences, Siksha ‘O’ Anusandhan (Deemed to be University), Bhubaneswar 751 003, India; amitkrnayak@yahoo.co.in; 3Department of Pharmaceutics, Royal School of Pharmacy, The Assam Royal Global University, Guwahati 781 035, India; prithvirajin@gmail.com; 4Department of Pharmaceutical Chemistry, Gupta College of Technological Sciences, Asansol 713 301, India; sabyasachibanerjee04@gmail.com; 5Department of Pharmaceutics, M.R. College of Pharmaceutical Sciences and Research, Balisha 743 234, India; talktobankim@gmail.com

**Keywords:** cancer, nanobiocomposites, nanotechnology, polymer, therapeutics

## Abstract

Cancer is one of the most common lethal diseases and the leading cause of mortality worldwide. Effective cancer treatment is a global problem, and subsequent advancements in nanomedicine are useful as substitute management for anti-cancer agents. Nanotechnology, which is gaining popularity, enables fast-expanding delivery methods in science for curing diseases in a site-specific approach, utilizing natural bioactive substances because several studies have established that natural plant-based bioactive compounds can improve the effectiveness of chemotherapy. Bioactive, in combination with nanotechnology, is an exceptionally alluring and recent development in the fight against cancer. Along with their nutritional advantages, natural bioactive chemicals may be used as chemotherapeutic medications to manage cancer. Alginate, starch, xanthan gum, pectin, guar gum, hyaluronic acid, gelatin, albumin, collagen, cellulose, chitosan, and other biopolymers have been employed successfully in the delivery of medicinal products to particular sites. Due to their biodegradability, natural polymeric nanobiocomposites have garnered much interest in developing novel anti-cancer drug delivery methods. There are several techniques to create biopolymer-based nanoparticle systems. However, these systems must be created in an affordable and environmentally sustainable way to be more readily available, selective, and less hazardous to increase treatment effectiveness. Thus, an extensive comprehension of the various facets and recent developments in natural polymeric nanobiocomposites utilized to deliver anti-cancer drugs is imperative. The present article provides an overview of the latest research and developments in natural polymeric nanobiocomposites, particularly emphasizing their applications in the controlled and targeted delivery of anti-cancer drugs.

## 1. Introduction

Extensive research efforts have been undertaken in material science engineering and technology to facilitate the development of diverse biomaterials for biomedical applications [1,2]. Nanotechnology research has identified a trendy and thriving study area focused on developing diverse nanomaterials for biomedical applications [3,4]. The fundamental cellular subunits and tissues exhibit clear division at the nanoscale level. Consequently, comprehending nanobiology, nanotechnology, and nanomaterials represents a novel frontier within the biomedical research and development domains [5]. Polymeric nanoparticles [6,7] and nanocapsules [8,9,10,11,12], magnetic nanoparticles [13,14], lipid nanoparticles [15,16,17], metallic nanoparticles [18,19,20,21,22,23], ceramic nanoparticles [24,25], nanogels [26], nanovesicles [27], nanotubes [28], and nanocomposites [29,30] are among the vital biomedical nanomaterials.

Nanocomposites are polyphasic materials characterized by at least one phase with nanoscopic dimensions in one, two, or three dimensions. Nanocomposites exhibit distinct characteristics compared to traditional composite materials, owing to the significantly greater surface-to-volume ratio of the reinforcing phase and its notably higher characteristic ratio. Additionally, they demonstrate enhanced ductility without any accompanying reduction in strength and heightened resistance to scratching [31]. Nanocomposites are widely utilized in various fields owing to their distinctive design capabilities, environmentally sustainable characteristics, facile manufacturability, and economic viability. Reinforcement materials improve the physical and mechanical properties of the matrix.

Reinforcement materials may consist of inorganic constituents such as metals, metal oxides, other inorganic components, fibers like electro-spun fibers, carbon nanotubes, and sheets like exfoliated clay stacks [32,33]. Various organic and organometallic compounds, natural and synthetic polymers, and other comparable materials are frequently combined with clays, fullerenes, inorganic nanoclusters, metals, oxides, or semiconductors to create nanocomposites [34]. The interactions between the nanofiller and the polymer matrix significantly influenced the properties of hybrid structures in nanocomposites [35]. The fabrication or synthesis of nanocomposites presents challenges, including the need to control fabrication/synthesis procedures, ensure compatibility of diverse material constituents, and achieve desirable and unique material characteristics [34,36]. As nanoengineered biomaterials, nanobiocomposites consist of different nano biomaterials incorporated into the bulk biomaterials, which are recently being researched and developed for various multi-functional biomedical uses, including drug delivery and targeting, tissue engineering, antimicrobial properties, stem cell therapy, wound dressings, cardiac prostheses, cancer therapy, biosensors, and enzyme immobilization [37,38,39].

In light of increasing environmental apprehensions regarding synthetic materials, there has been a recent inclination toward investigating natural materials and their utilization in various applications. Many naturally occurring biomaterials exhibit advantageous properties for specific biomedical applications. There is a growing interest in utilizing natural biopolymers, particularly in the biomedical domain, such as drug delivery and targeting. Numerous nanobiocomposites are employed for drug delivery and targeting in cancer therapy. Natural polymers are biodegradable and can be produced or extracted sustainably from renewable natural resources. Natural polymeric nanobiocomposites have been found to enhance drug delivery and targeting potential for anti-cancer therapy. Natural nanoformulation exhibits effective therapy, reducing drug accumulation in healthy tissues and minimizing associated side effects. In recent decades, many nanomaterials exhibited their potential applications in cancer therapeutics. The effective administration of anti-cancer medications to specific cancerous sites or cells has the potential to elicit the intended clinical outcome in the field of cancer therapeutics. In recent times, the utilization of nanobiocomposites composed of natural polymers to encapsulate several anti-cancer drugs has exhibited significant potential in regulating and directing the administration of such drugs for cancer treatment. In addition to controlled drug delivery, the ability to target specific cancerous cells, tissues, or organs is made possible by molecular interaction between anti-cancer drugs, entrapped polymeric nanobiocomposites, and the biological microenvironment. This interaction allows for efficient and effective targeted delivery of anti-cancer drugs.

This review article offers comprehensive, up-to-date knowledge on the expansion of different natural polymeric nanobiocomposites to deliver anti-cancer drugs. Topics such as cancer treatment, various approaches for drug delivery in cancer therapy, classifications of nanobiocomposites, processing of nanocomposites, and the importance of different natural polymeric nanobiocomposites for the delivery of anti-cancer drugs are discussed. This review discusses various types of natural polymeric nanobiocomposites, their hopeful features for cancer treatment, and their underlying mechanisms. We have also discussed the limitations, potential challenges, and future perspectives of natural polymeric nanobiocomposites for the delivery of anti-cancer drugs and for developing these nanocomposites for clinical application. We also highlighted the improvement in designing, developing, and optimizing these natural polymeric nanobiocomposites and their results on cancer therapy. Finally, the review summarizes the perception, the lessons learned so far, and the viewpoint toward further developments in this field.

## 2. Cancer

### 2.1. General Features

Cancer is a widely used term to describe diverse medical conditions characterized by uncontrolled cellular growth and proliferation originating from a specific site, which can affect multiple tissues throughout the body [40,41,42,43,44,45,46]. The global public health concern in question is of significant magnitude [47]. Cancer has emerged as a highly detrimental illness, with its incidence increasing unexpectedly. Annually, many individuals globally receive a cancer diagnosis, with over 50% of them succumbing to the disease. According to recent statistics, the United States experienced a notable rise in cancer incidence and mortality rates in 2020, with an estimated 1,806,590 new cases and 606,520 deaths [48]. Adopting behaviors and practices associated with elevated cancer risks, such as tobacco use, poor dietary choices, sedentary lifestyles, and alterations in reproductive patterns, increases cancer incidence [49,50]. The alarming statistics call for a collaborative effort among nations to promote healthy behaviors and advance scientific research toward innovative therapeutic solutions to eliminate the disease and enhance people’s well-being affected by this significant public health challenge.

### 2.2. Cancer Development

It is a traditional nomenclature for cancer encompassing all malignancies, causing social panic due to increased mortality and suffering. Cancer is associated with a multitude of factors that impact the regulation of pathways governing cellular proliferation and differentiation [51]. The malignant neoplasm exhibits a deficiency in cellular differentiation associated with impaired functionality and can exhibit a variable rate of progression. It exhibits locally aggressive behavior and can disseminate to other body regions via a limited population of tumor stem cells that can migrate to distant sites. The mechanism of carcinogenesis is a multi-stage, usually slow process associated with mutagenesis [52]. This mechanism passes through different phases of initiation, promotion, and progression, as shown in (Figure 1).

During the promotion phase, carcinogen (oncopromotors) exposed-initiated cells modify gene expression through long and continuous exposure. Prohibiting early exposure to the carcinogen can result in the carcinogenesis process coming to a halt [51,52].

Cell cycle checkpoints are an obstacle that the carcinogenesis process can eliminate. Cell cycle checkpoints maintained DNA integrity from the parent to the daughter cell. Carcinogenic agents play a role in the mutagenic and epigenetic processes that convert a healthy cell into a neoplasm [54,55].

### 2.3. Cancer Treatment

Systemic treatments (target therapy, hormone treatment, chemotherapy, and immunotherapy), local therapies (radiotherapy, surgery, and phototherapy) (Figure 2), or in combination, treat cancer. Cancer mortality has decreased in developed countries due to advancements in these therapies [56,57]. Surgery, radiotherapy, and chemotherapy are the three most commonly used therapies for malignant neoplasia. Recently, immunotherapy and photodynamic therapy have been employed as supplementary treatments in cancer management [58].

Though the metastasis process has not yet started, surgery is a viable option [59]. Chemotherapy may serve as adjuvant therapy after surgical intervention to eliminate any remaining cancerous cells and avert metastasis [60]. Approximately 50% of individuals diagnosed with cancer are mandated to receive radiation therapy at least once during their treatment [61]. Since the advent of the linear accelerator in 1960, radiotherapy has become a standard method for cancer treatment. However, eradicating cancer metastasis was not achieved [62]. Radiation destroys the tumor cells’ DNA, resulting in cellular death. The therapy’s effectiveness is restricted due to its potential to affect healthy cells [63,64].

Since the 1940s, cancer has been treated with conventional chemotherapy or cytotoxic agents [62]. Since then, several cytotoxic anti-cancer drugs have been produced, isolated, characterized, and commercialized. Currently, a variety of cytotoxic chemotherapeutic agents are accessible for use. These include alkylating agents such as temozolomide, cyclophosphamide, oxaliplatin, and cisplatin; antimetabolites like cytarabine, methotrexate, capecitabine, and fluorouracil; anti-tumor antibiotics such as epirubicin, doxorubicin, and bleomycin; topoisomerase inhibitors like etoposide and irinotecan; and microtubule stabilizers such as docetaxel, paclitaxel, colchicine, and vinca alkaloids [56].

Minchinton and Tannock [65] reported an eloquent review of the strategies to improve drug penetration through tumor mass and the design of selective compounds that have the targeted ability to penetrate tissue. Notably, 100 nm particles or larger generally do not penetrate well throughout the tumor mass, and smaller nanoparticles do not accumulate sufficiently in the tumor vasculature by the enhanced permeability and retention (EPR) effect and do not achieve good tumor penetration.

The size, shape, and surface modifications, which alter the pharmacokinetics and intracellular mechanisms, can be chemically modified to have a significant therapeutic impact in vivo. Investigations into the toxic effects following nanopolymer internalization are minimal, as many nanopolymers are inert delivery vehicles with little or no toxic effects when they release their contents [66].

The impact of nanopolymers’ physical characteristics on therapeutic effectiveness is under investigation. However, the consensus remains that nanoparticles’ effects and final properties in the endo-lysosomal vesicles of cells are unknown. For example, nanoparticles in the intracellular cytosol space can activate several biological functions, including disrupting mitochondrial function, eliciting the production of reactive oxygen species, and activating the oxidative stress-mediated signaling cascade. Other reports have demonstrated that hydrophilic titanium oxide TiO_2_ nanoparticles are oncogenic. Large nanoparticles do not extravasate far beyond the blood vessel, whereas small nanoparticles travel deep into the tumor but remain there only transiently. Therefore, optimizing the next generation of nanopolymers focusing on the intracellular therapeutic mechanisms after internalization is essential to successfully translate these drug-delivery systems to the clinic [67].

Several concerns arise regarding second-generation nanomaterials. First, there appears to be an overreliance on the EPR effect to deliver nanoparticles into the tumor, although this phenomenon may not be a universal property of all tumors in human patients. Second, no single nanoparticle size can access all tumor areas and accumulate in significant quantities. Large nanoparticles do not extravasate far beyond the blood vessel, and small nanoparticles travel deep into the tumor but remain there only transiently. Third, The barrier effect cancels out the benefits of active targeting, whereby most nanoparticles do not travel beyond the first few layers of cells because they adhere to their targeted receptors [68].

Targeted therapy employs specific molecules to selectively focus cancerous cells on their surfaces, inhibiting their growth and proliferation. As a consequence of genomic research, pertinent somatic mutations and epigenetic alterations have been discovered, which may assist in guiding the most efficient treatment, optimizing medication dose, and tracking the patient’s susceptibility to drug side effects [60].

### 2.4. Approaches for Drug Delivery in Cancer Therapy: General Considerations

Targeted drug delivery addresses many critical disadvantages of free medications used in conventional medicine, including very poor drug bioavailability, the requirement for higher dosages, and severe side effects [69]. Targeted drug distribution generally contributes to the selective and quantitative accumulation of drugs within target tissues and organs, regardless of the drug administration mechanism or pathway [70,71]. Moreover, the recognition of the objective can occur at various tiers, either within particular cells exclusive to the organ or even in distinct constituents of these cells, such as cell surface antigens [71]. The term “targeted treatment” or “targeted medication” pertains to the specific molecular interaction between a drug and its receptor. A high agent concentration is maintained at the disease’s affected site(s) to mitigate potential adverse effects, whereas its concentrations in non-target organs and tissues remain below specific thresholds [70,71].

Various targeting drug-delivery systems with unique characteristics, like enhanced drug encapsulation capability, improved drug stabilization, regulated drug release, and enhanced tumor aggregation through targeting modulation, have been thoroughly optimized and intensively studied due to the rapid advancement of innovative nanotechnology [69]. Nanoformulations must possess optimized surface charge, shape, and size that facilitate their prolonged circulation in the bloodstream before their elimination to enhance the efficacy of cancer therapy. Furthermore, it must meet the following requirements: (1) boost the pharmacokinetics and pharmacodynamics profile of chemotherapy drugs; (2) selectively destroy tumor cells without harming healthy cells using a regulated drug discharge at its active form; (3) enhance drug uptake and intracellular transport; and (4) minimize dose-limiting toxicity [72,73].

Without a doubt, the aim of targeting strategies has been to develop a successful antineoplastic therapy that does not damage normal tissues. With an improvement in the nanocarrier’s circulating time in the bloodstream, it is supposed to improve the potential of reaching cancerous cells or tumor tissue, thereby lowering adverse side effects and drug dosage [71,74].

Nanoparticles are suitable for encapsulating, protecting, and modulating therapeutic molecules in the blood supply and tissue distribution [75]. Passive and active targeting associated with vascular targeting, nuclear targeting, multi-stage targeting, and magnetic field targeting helps to accumulate nanoparticles within the tumors [76]. Figure 3 and Figure 4 illustrate how anti-cancer drugs are delivered to tumor sites via nanoparticles, utilizing two types of drug targeting.

#### 2.4.1. Passive Targeting

Compromised tumor vasculature and lymphatic drainage in the tumor microenvironment are exploited in passive targeting [78]. Because tumors increase vascular permeability and decrease lymphatic activity, biomacromolecules and nanosized drug-delivery systems have an easier time penetrating the capillary endothelium and entering the interstitial space, a phenomenon referred to as enhanced permeability and retention (EPR) [75,79]. As a result, drug release begins when nanoparticles extravasate from the tumor microvasculature, resulting in a build-up of therapeutic agents in the tumor interstitium. Encapsulating the active component into a nanoparticle or macromolecule helps to achieve passive targeting. Nanoparticles or macromolecules enter the target organ through the EPR mechanism [71,80]. Internalization in the cancer cell is critical as drug dispersal declines outside the cancer cell, effectively enhancing the medication’s therapeutic ability [81]. Due to this, passive targeting strategies are a compensatory process for the diffusion of nanoformulations in angiogenesis. It could also use nanoparticles’ inherent features to induce tumor targeting [81,82]. Figure 5 shows a schematic image of nanoparticle aggregation within a tumor induced by passive targeting.

High drug plasma concentrations for an extended period at the target locations are the requirements for passive transport and retention of nanoparticles in the tumor microenvironment [75,82]. Nanoparticles have low circulation half-lives due to their absorption by cancerous cells and extravasated constituents of the mononuclear phagocyte system (MPS). The macrophages’ affinity for the colloidal nanoparticles and the phagocytosis will restrict their movement into cancer cells, enhancing their persistence in the tumor interstitium [75]. Passive targeting, on the other hand, has the disadvantage of not being able to provide a high adequate drug dosage to the tumor microenvironment. The emphasis has increasingly changed from passive to active targeting nanoparticles due to this technology’s lack of tumor sensitivity and inability to regulate the release of entrapped agents [82].

#### 2.4.2. Active Targeting

In contrast to passive targeting, active targeting has higher precision. The approach leverages nanoparticles’ enhanced permeability and retention (EPR) effect, allowing them to selectively bind to target cells and accumulate at the tumor site by exploiting the differences in receptor expression between malignant and non-malignant cells [82,83]. Cancer cells exhibit overexpression of surface receptors that possess the potential for targeted intervention. Furthermore, each cell exhibits a unique characteristic that can be effectively utilized for successful recognition of the cell [71]. Active targeting involves the utilization of overexpressed tumor-specific antigens or receptors to guide drugs into tumor tissue. It activates cancer-specific receptors on the surface of tumor cells, which contain ligands or antibodies to these proteins. Receptor-mediated endocytosis delivers the medication payloads within the tumor tissue [82,84]. Therefore, “active targeting” essentially refers to a particular “ligand-receptor type interaction” with intracellular localization that takes place just after blood circulation and extravasation [70,82,84]. PEGylation (modifying the surface of nanoparticles with polyethylene glycol) prolongs blood supply, and enhancing the EPR effect should improve delivery to the tumor location [70]. Figure 5 depicts a schematic image of nanoparticle aggregation inside a tumor as a prerequisite for active targeting.

**Figure 5 pharmaceutics-15-02064-f005:**
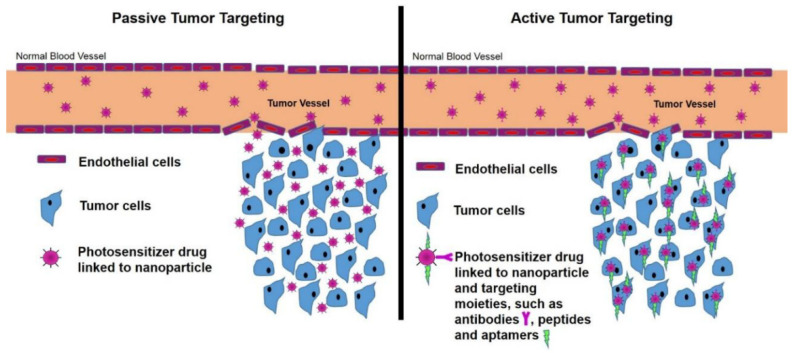
Schematic image of nanoparticle aggregation inside a tumor as a prerequisite for passive and active targeting (reused from Kruger and Abrahamse, 2018, distributed under Creative Commons Attribution 4.0 International License, which permits unrestricted use, distribution, and reproduction in any medium. http://creativecommons.org/licenses/by/4.0/ (accessed on 3 May 2023)) [85].

Active targeting addresses specifically defined areas with reduced side effects. The existence of a tumor-targeting ligand does not lead to elevated nanoparticle aggregation in tumors [75]. An “active strategy” does not always imply successful delivery to the whole tumor. As a result, both passive and active approaches can be used in tandem to get the most potential nano-designed “magic bullets” [70,71,75].

### 2.5. Nanoscale Drug-Delivery Systems for Treating Cancer

Drug delivery emerged as one of the initial domains of cancer research to undergo expansion. Upon administration, the drug is gradually released into the patient’s circulatory system, resulting in advantageous outcomes such as diminished toxicity and adverse reactions and enhanced therapeutic effectiveness [86]. Scientists are attempting to create new carriers for anti-cancer drug delivery directly targeted at cancer tumors in light of this dire health situation [87].

Nanoparticles have demonstrated significant influence as an oncology therapy modality in recent decades. Nanoparticles have the potential to enhance drug delivery to cancer cells and improve their destruction by concentrating drugs in specific regions of the body. However, the efficacy of this approach is contingent upon the tumor form and stage of development [88]. Nanoparticles utilizing passive or active targeting mechanisms increase the intracellular amount of drugs in cancer cells [89]. These processes have utilized various materials, either alone or in combination. These materials are lipids (solid lipid nanoparticles, liposomes), polymers (nanoparticles, micelles, dendrimers), and metals [90]. Polymer-based technologies for the controlled release of anti-cancer agents have been developed over the last few decades [91].

Polymeric nanoparticles, which have diameters smaller than 1 µm, serve as carrier models and can be produced using various techniques that depend on the type of medication, encapsulation, and their intended use [92]. One of the primary advantages of utilizing dosage forms containing nanoparticles is their subcellular dimensions, which have the potential to facilitate optimal drug accumulation at specific target locations [93]. Nanomaterials have garnered significant attention as potential drug-delivery systems due to their capacity for controlled and sustained drug release and their ability to disperse a diverse array of therapeutic agents [93]. Polymeric nanoparticles have been identified as highly effective nanocarriers for cancer therapy because they efficiently eliminate cancer cells and exhibit a prolonged systemic circulating half-life, thereby enhancing therapeutic efficacy [88,91]. The nanoparticles possess a considerable surface modification and functionalization capacity by attaching diverse ligands. These properties enable the facilitation of active targeting. Nanoparticles with ligands on their surfaces enhance their selective binding to overexpressed receptors on target cells. This approach confers benefits by enabling the escalation of drug concentration at specific sites while concurrently mitigating drug exposure to healthy cells [94]. Several studies have also shown that these ligands could be used to actively treat cancer [88,94]. For instance, in an in vitro study, conjugating PLGA with transferrin equipped with bortezomib improved the target penetration of PGLA nanoparticles into pancreatic cancerous cells [95]. Surface improvements of cisplatin-charged nanoparticles with ligands such as the F3 peptide resulted in anti-angiogenesis tumor regression, which could help treat people with ovarian cancer [96]. The optimal polymer nanoparticulate system for cancer treatment can effectively regulate its superficial properties and its particle size for the control of infiltration, improve permeation, versatility, solubility, and release therapeutic agents, ensuring that the specific activity and target area are achieved on schedule and at the desired rate [88,97].

## 3. Definition and Classifications of Nanobiocomposites

In material science and biomedical engineering, the term “composite” refers to a substance that is not naturally occurring and is composed of two or more materials with distinct chemical and physical properties. The constructed composite has a unique set of properties that differ from those of its components. Composite materials typically consist of two distinct phases: a continuous bulk phase called the matrix and a non-continuous phase known as the reinforcement. The matrix in the bulk phase exhibits a lower Young’s modulus and more remarkable elastomeric behavior, whereas the reinforcement displays superior load capacity and physical and mechanical characteristics. In biomedical engineering, it is customary for the composite matrix to consist of a biopolymer, while an inorganic bioactive substance is a reinforcement. These composites are called “biocomposites” and have particular uses in biomedical engineering [98].

Nanocomposites are a type of nanoengineered synthesized substance. Nanocomposites are multiphase, rigid nanostructured entities that integrate multiple layers of various materials, including at least one phase with one, two, or three dimensions in the nanoscale range [99]. Different fortified materials strengthen nanocomposite matrices’ physical, physiochemical, biomedical, and mechanical properties. Figure 6 exhibited a scheme of nanocomposites’ conceivable configuration. Nanocomposites are distinguished from conventional composites by a higher surface-to-volume ratio for the reinforcing phase, increased ductility without lowering the product strength profile, and scratch resistance [31]. Nanocomposites exhibit distinct properties due to their high surface-to-volume ratio compared to their larger-sized counterparts. It also alters the way nanoparticles interact with bulk materials.

Consequently, the composite will outperform its constituent parts by a significant margin. The durability of nanocomposite materials was significantly more excellent than that of bulk component materials, with a difference of up to 1000 times. Reinforcing materials such as particles, sheets, or fibers can be utilized. Compared to conventional composite materials, the interface region within the matrix and reinforcement phase typically exhibits a significantly greater magnitude. Various types of nanocomposites have been studied and established over the last few decades. Natural polymers, semi-synthetic polymers, synthetic polymers, metals and their oxides, structured carbons, ceramics, and other inorganic materials make up most of such nanocomposites [100].

Nanobiocomposites, nano biocomposites, and biomedical nanocomposites are all used to describe nanocomposites engineered for various biomedical applications. In general, based on matrix-reinforcement composition, nanobiocomposites are categorized as organic–organic nanobiocomposites (e.g., polymer–polymer nanobiocomposites), organic–inorganic nanobiocomposites (e.g., polymer–ceramic and polymer–metal nanobiocomposites) and inorganic–inorganic nanobiocomposites (e.g., ceramic–ceramic, metal–metal, and ceramic–metal nanobiocomposites). Nanobiocomposites have a wide range of applications. Drug distribution, antimicrobial activity, wound dressings, tissue engineering, cancer treatment, stem cell therapy, cardiac prostheses, bioengineered blood vessels, enzyme immobilization, biosensors, and other bio-medical applications of nanobiocomposites are among the most popular [101].

## 4. Manufacturing/Processing of Nanocomposites

The same techniques used to create standard composites are also employed to produce nanocomposites. In situ polymerization, melt mixing, melt intercalation, solution mixing, extrusion, precipitation, spray mixing, and sol–gel processes are only a few methods used to create new materials [102]. Hand lay-up and spray-up methods create layered composite structures using natural fibers. Furthermore, many novel technologies have effectively produced nanocomposites utilizing sustainable resources. These techniques include electrospinning and direct dispersion of nanofibers.

However, academic research is more frequent than industrial development to use specific approaches. The industrial production sector tends to prioritize simplified and streamlined manufacturing processes. Specific methodologies incorporate the utilization of sustainable resources. Modifying starch, gelatin, gluten, cellulose, and chitosan can yield sustainable polymers and composites derived from bio-renewable sources [103]. Thus, the fundamental attributes of natural polymers, including their bio-degradability, bio-compatibility, and facile processability, are conveyed into the resultant composites. Renewable materials, including agricultural waste or biomass, have been utilized to obtain natural polymers, which are abundant [104,105]. Polymeric nanocomposites have been extensively produced in the past decade using techniques such as extrusion, melt intercalation, and in situ polymerization.

### 4.1. Solution Mixing

Academic scientific research utilizes this method. The matrix material is typically dissolved within a solvent and blended with the additive. Subsequently, coating, molding, and film casting techniques generate the composite frame. Several limitations associated with this procedure encompass the need for a substantial quantity of solvent, its hazardous nature, and the intricacies that ensue during the elimination of the solvent after composite formation [102].

### 4.2. Melt Mixing

The initial step of this procedure involves the transformation of the matrix substance into a liquid state by applying heat beyond its melting threshold. Regulating the assimilation of the additive substance into the liquid melt enables the composite material to undergo molding or shaping processes. The procedure commonly employs matrices of fusible substances, such as gelatin, derived from renewable sources. Meanwhile, additives are renewable materials. Structurally renewable fiber is preferable for producing particulate or layered additive composites [106].

### 4.3. Extrusion

The extrusion process is the most economically efficient and preferred method for producing composites and nanocomposites based on thermoplastics [102]. The extrusion process involves the melting of raw materials and their subsequent introduction into a melting channel, to be shaped into the desired form using one or more screws within the channel’s outlet block. Nanoscale additives are added to the molten raw material in manufacturing to create nanocomposites. The injection molding technique produces specific composites of polypropylene reinforced with natural fibers. Unsaturated polyester, vinyl ester resins, and epoxy produce such composites. Extrusion and injection molding facilitates large-scale production with convenience. However, these techniques are solely applicable to the production of thermoplastic composites. The proliferation of technical products has ensued due to escalated production [107].

### 4.4. In Situ Polymerization

An in situ polymerization technique involves polymerizing the additive’s monomer structure. The additive can diffuse through the matrix when it is present during polymerization. The polymerization process of commonly utilized organic monomers can convert sustainable additives into composite materials, including chitin, cellulose, lignin, and starch. The process of synthesizing composites using monomers derived from renewable sources involves polymerization, which is a method that utilizes renewable sources. Combining nanostructures such as carbon nanotubes (CNTs), clay, and graphene with terpenes and terpenoids created nanocomposites during polymerization. Another illustration pertains to using prevalent additives to generate nanocomposites derived from double-bonded or epoxidized monomers procured from vegetable oils. The in situ polymerization process fabricates composites while ensuring minimal additive agglomeration. Consequently, the uniformity of the generated materials exhibits a high degree of homogeneity [106].

### 4.5. Sol–Gel Process

The sol–gel method involves the hydrolysis of a metal alkoxide followed by its condensation [108]. This method incorporates metal oxide nanoparticles, such as zinc oxide (ZnO), titanium dioxide (TiO_2_), silicon dioxide (SiO_2_), and other similar materials, into ceramic matrices. Thus, this approach synthesizes inorganic metal oxide additives in manufacturing sustainable nanocomposites. In contrast, using metal requires the implementation of expensive substances, such as alkoxides, that are resistant to moisture. The sol–gel technique and hybrid composite preparation processes provide potential solutions to the mechanical strength limitations inherent in bionanocomposites. However, these methods are still in the nascent stages of development and have yet to be widely adopted by industry. A responsive molecule is affixed to the surface of the matrix to improve mechanical properties and reduce phase discrepancies between the additive and the matrix material. Looking ahead, the notion of bonding is a highly stimulating and auspicious prospect due to its robust covalent association with the matrix framework [102].

### 4.6. Electrospinning

Electrospinning is a cost-effective, precise, swift, and secure technique for generating diverse nanofibers (derived from synthetic or natural polymers). These nanofibers exhibit diameters that span from the micrometer to the nanometer scale [109]. The electrospinning method for synthesizing nanofibers, mainly those containing polymers, is considered the most advantageous technique because it offers diverse options for producing biomaterials. This approach yields a substantial quantity of nanofibers that are extensively employed. Nanofibrous polymer composites produced through electrospinning have gained significant attention due to their exceptional characteristics, such as high aspect ratios, surface area volume ratios, variable pore sizes, and oxygen permeability. These characteristics make them ideal for many applications, including electro-spun fiber nonwoven mats, dressings, and targeted drug delivery. This approach involves dissolving the polymer solution with the required characteristics in a suitable solvent and inserting it into the syringe. Following this first phase, a voltage at a high potential ranging from 5 to 50 kV is applied to the nozzle of the syringe to create an electrical field. Due to the surface tension force, the polymer solution takes a hemispherical shape as it exits the syringe needle. When the potential difference surpasses a specific threshold value, the spherical polymer droplet transforms into a conical shape and expands, forming a fiber. A collector with conductive properties gathers the produced fibers. This approach creates numerous nanocomposites using bio-renewable materials, such as wound dressings, medication delivery systems, and tissue support systems [102].

### 4.7. Resin Injection Methods or Resin Transfer Molding

Resin transfer molding or resin injection prepares nanocomposites. The methodology utilizes a dual-component cast with a manually operated inclination mechanism, expediting the procedure and guaranteeing prolonged durability. The components that strengthen a structure include felt fabric. The initial step in filling the mold involves inserting the reinforcing material, followed by the closure of the mold. The matrix is fortified with latex resins to avert the infiltration of fibers into the mold. The injection of resin into the mold under pressure is a laborious and time-intensive procedure. This technique involves the utilization of a vacuum to remove the air inside and ensure deep infiltration of the resin into the fibers [102].

### 4.8. Spray-Up and Hand Lay-Up Methods

Along with the above cutting-edge methods for creating composites derived from sustainable sources, conventional techniques like spray-up and hand lay-up are also used [102]. The hand lay-up technique is a rudimentary yet effective method for fabricating bio-based composites derived from sustainable resources. The present technique enables the production of composites that incorporate additives up to 30%. The present methodology involves the incorporation of bio-derived additives into a pre-existing mold, followed by applying matrix resin onto the fiber using rudimentary manual instruments such as a brush. The hand lay-up technique involves the deposition of a fiber or fiber-structured additive onto a mold, followed by impregnation with liquid resin. The material has the desired width with the repeated process mentioned above. Polyester, epoxy resins, vinyl ester, and phenolic resins use this methodology. Low-volume manufacturing preferred the hand lay-up technique. This methodology utilizes resins derived from sustainable sources. Numerous composites have been fabricated, particularly utilizing resins derived from vegetable oils. The mechanized forms of hand lay-up, namely mix coating and spray-mixing (spray-up) techniques, are employed to acquire various products. Forming a mold involves the application of a mixture of chopped fibers and liquid resin onto an open mold using a spray gun. Typically, in these methods, the application of natural fibers and resins occurs concomitantly through spraying onto or into the mold. The application-specific spray gun applies the resin mixed with the curing agent onto the surface of the container, along with the additive phase, in the form of fibers, nanoplates, particles, or fabrics. Upon spray application, the surface is subjected to a drying process, resulting in the acquisition of a composite structure [110] (Figure 7).

### 4.9. Other Methods

Various techniques can produce nanocomposites, including centrifugal casting, pultrusion, and nanofiber direct dispersion [102].

### 4.10. Rational for the Uses of Natural Polymers to Construct Nanobiocomposites

Polymeric nanocomposites present many possibilities in various fields, such as tissue engineering, nanocarriers, sensors, and antimicrobials. Polymeric nanocomposites exhibit significantly enhanced properties compared to their pure polymer counterparts, rendering them highly effective drug carriers [112]. Incorporating nanoparticles into the biopolymeric matrix can mitigate burst drug release, augment drug stability, and facilitate a more gradual and prolonged drug release [113]. Numerous studies have investigated the drug release characteristics of a wide range of nanocomposites incorporating various nanofillers, including but not limited to gold, silver, TiO_2_, ZnO, clay, silica, carbon quantum dots, graphene, carbon nanotubes, hydroxyapatite, core-shell, bio-glass, yolk-shell, and coupled/impregnated nanostructures [114,115,116,117,118].

Polymeric nanocomposites exhibit considerable potential in disease theranostics, while nanotechnology holds great promise in developing cancer drug carriers. Using biodegradable polymer-based nanocomposites loaded with anti-cancer agents is advantageous for controlled and targeted drug delivery due to their ability to potentially elevate the drug concentration in cancerous tissues, thereby improving the antitumor efficacy. The application of nanocomposites in cancer therapy entails incorporating pharmaceutical agents and modifying nanoparticles with biomolecules to facilitate interaction with the tumor. When the matrix exhibits hydrophilic properties as a polymer, it augments the solubility of the nanoparticle, thereby enhancing its compatibility [105]. The diagram depicted in Figure 8 demonstrates the amalgamation of various applications that possess the potential to serve as drug delivery agents and cancer cell sensors (Figure 8).

Drug-delivery carriers frequently comprise natural biopolymers, synthetic polymers, and polymeric nanocomposites. The utilization of natural polymers in biotechnological and bio-medical applications has garnered attention due to their potential as viable options. These applications include diagnosis, bioactive therapy, and controlled drug delivery/targeting applications. The benefits of natural polymers stem from their biodegradability, biocompatibility, inherently low immunogenicity, and unique bioactive properties [120,121]. Recently, polymers from natural resources, such as animals, plants, seaweeds, and microorganisms, have been isolated [122]. Due to their surrounding by different kinds of functional groups or backbones, natural polymers exhibit physicochemically and structurally diverse materials [123,124]. These also reproduce some fortunate properties, such as being biodegradable, non-toxic, water-soluble, and having higher swelling capacity through simple chemical changes and stability [124]. These natural polymers also contain various chemical structures, i.e., functional groups or backbones, which may be effective sites for chemical interchanges [125]. These polymers, known as gels, show various three-dimensional interlinked molecular networking structures. These natural polymers’ gel strengths depend on some variables, including their molecular structure, the concentration of the polymer, and other variables, including pH, temperatures, and ionic strengths [126]. These characteristics have modified natural polymeric substances for utilization as raw materials for biomedical applications. In the biomedical field, natural polymers degrade into physiological metabolites, making them excellent and valuable materials for various applications, such as drug delivery [127]. During the past few decades, investigating these natural polymers for developing different kinds of nanobiocomposites has been proceeding consistently in diversified biomedical applications, including drug delivery [120]. Natural polymeric nanobiocomposites have emerged as a potential carrier system for targeted anti-cancer drug delivery due to their enhanced physicochemical, physical, bio-medical, and mechanical properties. Furthermore, the enhanced drug loading capacity and incorporation of nanoscale reinforcement within natural polymeric nanobiocomposites have expanded the horizons of drug delivery research [120].

## 5. Different Natural Polymeric Nanobiocomposites for Delivery of Anti-Cancer Drugs

Biodegradable natural polymers have been widely considered superior for encapsulating and administering various drug varieties. There is a significant focus on the pharmaceutical industry in numerous research studies. Polymers address the constraints associated with traditional dosage forms that possess improved bioavailability, biocompatibility, and reduced toxicity, thereby facilitating tailored and targeted drug-delivery techniques. Nanomedicines are an innovative form of pharmaceuticals. From a clinical perspective, polymers that are not biodegradable administer antibodies via local injection.

The category of non-biodegradable polymers encompasses various types of polymers, including but not limited to acrylic polymers, cellulose derivatives, and silicons. Polymethyl methacrylate (PMMA) is a polymer composed of non-biodegradable acrylic-based materials used as bone cement for implantation purposes and as PMMA beads [120].

### 5.1. Chitosan-Based Nanobiocomposites

Chitosan is a natural polysaccharide obtained by the partial deacetylation of chitin, a well-known marine-derived biopolymer that usually occurs as a significant component of crustacean exoskeleton and fungi cell wall [128]. Chitosan molecules possess a cationic charge and are composed of α-1, 4-linked 2-amino-2-deoxy-*α*-D-glucose (N-acetyl glucosamine) (Figure 9) [129].

It possesses biodegradable and biocompatible characteristics. Due to its possession of these two essential biomaterial properties, chitosan has many biomedical applications. Chitosan also possesses an intrinsic antibacterial character. It is a notified as “Generally Recognized as Safe” (GRAS) material by the US FDA. It is a biopolymeric excipient in the formulations of various dosage forms [130]. Figure 10 depicts a visual representation of a nanocomposite based on chitosan and incorporating various nanofillers. In recent decades, researchers have developed numerous chitosan-based nanocomposites by incorporating other biopolymers and bio-inorganics to enhance their efficacy in delivering anti-cancer drugs.

Abdel-Bary et al. [131] in the year 2020 fabricated chitosan-based nano-composites of cisplatin. Inorganic nanoparticles like magnetite, silicon dioxide, and graphene oxide functionalized the chitosan. They prepared different chitosan-based polymeric–inorganic nanocomposites (namely chitosan-coated magnetite and silicon dioxide; chitosan-coated magnetite, silicon dioxide, and graphene oxide; chitosan-coated silicon dioxide and chitosan-coated magnetite). The chitosan-coated magnetite nanocomposites showed 87% cisplatin loading, whereas the chitosan-coated magnetite, silicon dioxide, and graphene oxide nanocomposites showed 84% cisplatin loading. The chitosan-coated magnetite nanocomposites exhibited the highest cisplatin release (91%) in in vitro drug release evaluation. Chitosan-polypyrrole nanocomposites (CS-PPy NCs) improved the individual components’ biocompatibility, stability, conductivity, and strong near-infrared (NIR) absorbance. CS-PPy NCs also demonstrated cytotoxic activity against MDA-MB-231 cells with reduced cell viability, although the percentage of cell viability is above 60% at a concentration of 500 µg/mL, indicating low cytotoxicity and good biocompatibility in nature. MDA-MB-231 carcinoma cells were all killed on exposure to CS-PPy NCs incubation and 808-nm NIR laser irradiation with a power density of 2.0 W/cm^2^. It further induces cell late apoptosis or necrosis. Laser irradiation elevated the process. In vivo studies revealed that the CS-PPy NCs incubation and NIR laser irradiation reduced the tumor volume and completely eradicated the tumors in tumor-bearing female BALB/c nude mice with a dose of 100 μL of 5 mg/mL [132].

In a separate study, a selenium-coated chitosan-stabilized iron oxide nanocomposite was synthesized and exhibited cytotoxic activity against MB-231 breast cancer cells with a decrease in cell viability in a concentration-dependent manner. It reduced the cell viability to 40.5% at a concentration of 1 µg/mL [133].

The 17*β*-hydroxy-17*α*-picolyl-androst-5-en-3*β*-yl-acetate is an androstane commonly used as an anti-cancer drug against various cancer types. A nanocomposite was prepared by entrapping the drug in carrier hydroxyapatite (HAp) coated with chitosan (Ch)-poly(lactic-co-glycolic acid) (PLGA) polymer blends (Ch-PLGA) which improved the drug efficacy. HAp/Ch-PLGA spherical nanocomposites loaded with 17*β*-hydroxy-17*α*-picolyl-androst-5-en-3*β*-yl-acetate induced three times more profound cytotoxicity towards A549 lung cancer cells than human lung fibroblasts (MRC5 cell line) [134].

Nejadshafiee et al. (2019) prepared amine-functionalized Fe_3_O_4_ magnetic nanoparticles. The functionalized nanomaterials were coated with chitosan to fabricate Fe_3_O_4_@metal–organic frameworks (MOFs) folic acid nanocomposite systems, a drug delivery carrier. Folic acid (FA) and the amine group of chitosan were conjugated. Curcumin/5-FU-loaded Fe_3_O_4_@Bio-MOF-FC nanocomposites exhibited low toxicities compared to free curcumin and 5-fluorouracil. It further exhibited high cytotoxicity against MDA-MB-231 cells (epithelial-like breast cancer cell line), evident from the decrease in percentage cell viability in the concentration range of 0–100 µg/mL [135].

In vivo T2-weighted MR imaging of tumor-bearing BALB/c mice utilizing a 3.0 T human MRI scanner revealed that the Fe_3_O_4_@Bio-MOF-FCNCs treatment reduced the signal intensity value with negative T2 contrast enhancement associated with higher absorption of the nanocomposites in the tumor cells than normal cells [135].

Later on, chitosan was grafted with poly (2-amino thiophenol) (P2-ATH) by chemical oxidative polymerization, and its nanocomposite (chitosan-gr-p2-ATH with AgNPs) was synthesized by assembling with silver nanoparticles. X-ray diffraction, scanning electron microscope analysis, transmission electron microscopic analysis, and thermogravimetric analysis characterized the nanoformulation. It exhibited potent anti-cancer activity against HepG2 human liver cancer cell line in the concentration range of 0.977–2000 μg/mL, having an IC_50_ value of 11.22 μg/mL than chitosan alone which showed an IC_50_ value of 668.52 μg/mL [136].

Yang et al. (2017) employed a sol–gel method with slight modifications to fabricate a chitosan-based mesoporous magnetic nanocomposite utilizing silica (SiO_2_) and photosensitizer chlorin e6 (Ce6) to develop a controlled release of doxorubicin, an anti-cancer drug, in cancer therapeutics. Chitosan is a blocking excipient in developing a mesoporous magnetic nanocomposite based on chitosan. The purpose of this excipient was to prevent the premature release of doxorubicin. Doxorubicin released from magnetic nanocomposite systems depends on pH. The nanocomposite demonstrated heightened antiproliferative efficacy against the MCF-7 breast cancer cell line, the EMT-6 syngeneic breast carcinoma cell line, and the MCF-7/ADR multi-drug-resistant breast cancer cells through the induction of apoptosis, which was attributed to the synergistic influence of other constituents present in the nanocomposite [137].

In vivo studies revealed that it further exhibited antitumor activity in EMT-6 tumor-bearing Balb/c female mice by reducing the tumor volume, but the doses were not specified. In vivo T2-weighted MR imaging of tumor-bearing BALB/c mice revealed negative T2 contrast enhancement in the tumor sites [137].

A similar study evaluated cisplatin release from different poly oxalates cross-linked chitosan (CS) nanocomposites. The ionic gelation technique encapsulates cisplatin on CS cross-linked with oxalic acid/succinic acid/citric acid/tartaric acid-ethylene glycol carriers. All these nanocomposites displayed cytotoxicity against MCF-7 cells by inhibiting their proliferation in a time- and concentration-dependent manner, having an IC_50_ value of 45 µg/mL. These nanocomposites showed significant bio-compatibility in vitro [138].

Biocompatibility, biodegradability, and drug-delivery properties of chitosan can improve the anti-cancer properties of rutin against the human lung cancer A549 cell line by synthesizing a Ch-CuO nanocomposite with rutin. The nanocomposite showed elevated cytotoxicity against the carcinoma cell line by inhibiting proliferation and inducing apoptosis, having an IC_50_ value of 20 µg/mL compared to rutin alone, which showed an IC_50_ value of 98 µg/mL [139].

N, N, N-trimethyl chitosan chloride (TMC), a chitosan derivative, is a biodegradable biopolymer with reducing and stabilizing properties. Silver nanocomposites were synthesized utilizing TMC and produced sphere-shaped particles with 11 to 17.5 nm diameters. TMC/Ag nanocomposite displayed cytotoxicity against lung carcinoma cells (A-549) and normal lung cells (WI 38), having an IC_50_ value of 12.3 μg/mL and 357.2 μg/mL, respectively [140].

In another experiment, Prabha and Raj (2016) developed polymeric nanocomposites comprising chitosan–polyethylene glycol–polyvinyl pyrrolidone to deliver curcumin in a targeted manner while also conjugating magnetic nanoparticles (Fe_3_O_4_). The anti-cancer efficacy of the drug-delivery system was evaluated in Caco-2 and HCT-116 cell lines, indicating a significant increase in activity relative to nonmagnetic nanoparticles [141].

Pan et al. (2017) created nanocomposites of graphene oxide functionalized with lactobionic acid and carboxymethyl chitosan. These nanocomposites deliver DOX in a targeted manner. The present study involved the adsorption-based loading of DOX onto the composites. The release behavior’s pH sensitivity is notable in the lactobionic acid-functionalized and LA-free materials. The findings suggest that the modified graphene oxides exhibit notable biocompatibility with the SMMC-7721 liver cancer cell line. However, upon loading with DOX and subsequent 24 h incubation, the modified graphene oxides may elicit cell death. For a 2 h incubation period, they examined the selectivity of graphene oxide composites. The results indicate that the DOX-loaded system exhibited toxicity towards the non-cancerous L929 cell line. However, the composite containing lactobionic acid demonstrated the capacity to selectively induce cell death in cancerous (SMMC-7721) cells, whereas the lactobionic acid-free counterpart did not exhibit any toxicity. The results of this study indicate that the altered graphene oxide substances exhibit considerable promise as viable contenders for targeted delivery systems of anti-cancer drugs [69].

Rasoulzadehzali and Namazi (2018) have produced pH-sensitive nano biocomposite hydrogel beads utilizing chitosan and graphene oxide-silver nanohybrid particles to facilitate the controlled release of DOX. UV-Vis spectroscopy analyzed the significant loading efficiency of DOX into the test beads. Furthermore, an assessment was conducted on the antibacterial efficacy, swelling behavior, and drug release kinetics of the synthesized nanocomposite beads. The study revealed that an elevated content of graphene oxide-silver nanohybrid particles led to an improved drug release profile in chitosan/graphene oxide-silver nanocomposite hydrogel beads, which was more consistent and regulated [142].

In a study, Xie et al. (2019) produced nanocomposites of chitosan/sodium alginate functionalized magnetic graphene oxide loaded with DOX for targeted delivery and photothermal therapy in cancer treatment. They synthesized magnetic iron oxide nanoparticles loaded onto graphene oxide nanosheets, then prepared chitosan-based nanocomposites using layer-by-layer self-assembly techniques. These chitosan/sodium alginate functionalized magnetic graphene oxide nanocomposites were 0.5 μm in diameter and were 40–60 nm thick. These also showed a superparamagnetic character. The findings of the stability analysis indicate a reduction in the agglomeration of chitosan-derived superparamagnetic nanocomposites. However, their stability increased in the biological medium. Electrostatic and π–π stacking interactions achieved DOX loading onto the nanocomposites. Chitosan and sodium alginate elevated the loading of DOX on the nanocomposites of magnetic graphene oxide. The study’s findings indicate that the chitosan-based superparamagnetic nanocomposites exhibit a pH-responsive behavior in releasing DOX in vitro. The drug release characteristics of DOX-containing nanocomposites, mGO-CS/SA-DOX, were enhanced for dispersion and pH sensitivity. The results of the cellular experiments demonstrated that mGO-CS/SA exhibited magnetically targeted cellular uptake and remarkable photothermal effects. Additionally, the toxicity of mGO-CS/SA-DOX was found to be concentration-dependent [143].

A stimuli-responsive hydrogel nanocomposite was synthesized in a separate study through the co-polymerization reaction of acrylic acid and N-isopropyl acrylamide on chitosan, followed by the in situ synthesis of Fe_3_O_4_ magnetic nanoparticles. The hydrogel nanocomposite was employed as a drug-delivery system (DDS) to achieve controlled release of the anti-cancer drug DOX. The study demonstrated that the nanocomposite exhibited a drug-loading efficiency of up to 89%. Additionally, in a sustained manner, the drug DOX was released during in vitro experiments. Furthermore, the nanocomposite exhibited dual pH and temperature responsiveness, with the release of 82% DOX from the hydrogel within a period of 48 h. Therefore, the nanocomposite based on CS exhibits potential as a carrier for controlled and sustained drug delivery [144].

In a study by Dhanavel et al. (2017), the synthesized nanocomposite used chitosan and palladium. Curcumin and 5-fluorouracil were encapsulated and used to treat colon cancer. The researchers adopted a straightforward, adaptable, and financially advantageous approach. The nanocomposite exhibited significant cytotoxicity against HT-29 cells, wherein a noteworthy inhibitory effect was observed upon concurrent administration of curcumin and 5-fluorouracil [145].

Hosseinzadeh et al. (2019) succeeded in developing a hydrogel nanocomposite that is responsive to stimuli. They utilized surface reversible addition–fragmentation chain transfer (RAFT) copolymerization of acrylic acid and N isopropyl acrylamide onto chitosan, followed by the in situ synthesis of magnetic nanoparticles. The hydrogel nanocomposite was modified and utilized as a proficient vehicle for the controlled discharge of DOX. According to the findings, the highest loading efficiency of DOX in the nanocomposite was 89%. The results of in vitro drug release studies indicated that the nanocomposites exhibited sustained-release characteristics for DOX. The nanocomposites’ dual temperature and pH responsiveness was demonstrated through in vitro release studies, revealing that the hydrogel released 82% of the total DOX within 48 h. The chitosan-based nanocomposite exhibits distinctive structures and properties that render it a viable candidate for serving as a drug carrier with the potential for controlled and sustained release of anti-cancer agents [144].

Bao et al. (2011) synthesized chitosan-functionalized graphene oxide nanoparticles containing camptothecin, a water-soluble anti-cancer medication. These nanoparticles exhibited improved water solubility and bio-compatibility of chitosan-functionalized graphene oxide. Chitosan-functionalized graphene oxide consists of an amide linkage between chitosan and graphene oxide. The loading of camptothecin into the nanocarriers involves a combination of π–π stacking and hydrophobic interactions facilitated by the functionalization of graphene oxide with chitosan. The nanocarriers demonstrated a regulated discharge of 17.5% camptothecin within a 72 h timeframe, as per the simulated conditions [146].

Parida and colleagues (2011) developed a new type of composite carrier by combining chitosan-polyvinyl alcohol with cloisite^®^ 30B to regulate the discharge of curcumin. The amalgamation of chitosan and polyvinyl alcohol polymers enhanced the tensile strength and flexibility of the films. The presence of chitosan in the formulation influenced the in vitro drug release in a pH-dependent manner. The alkaline release medium exhibited a more pronounced drug release (in vitro). Electrospun nanofibrous scaffolds composed of polyethylene, chitosan, and graphene oxide help in the controlled release of DOX. The liberation of DOX is primarily contingent upon the π–π stacking interaction between the drug and the graphene oxide nanofibers. Additional factors that contribute to this process include hydrogen bonding, electrostatic interaction, and the diffusion of DOX from the synthesized nanofibers [147].

Alginate–chitosan–pluronic composite nanoparticles deliver curcumin. The study determined that incorporating pluronic in the composite nanoparticle preparation formula increased the curcumin encapsulation efficiency of the chitosan-based composite nanoparticles. The biocompatibility of composite nanoparticles was demonstrated through experimentation on Hela cells, indicating the potential utility of alginate–chitosan–pluronic composite nanoparticles in administering hydrophobic anti-cancer drugs for cancer treatment [148].

The study conducted by Nanda et al. (2011) involved the development of chitosan-polylactide nanocomposites through the utilization of cloisite^®^ 30B. The research team also examined the potential of these nanocomposites to release paclitaxel, an anti-cancer drug. The findings of the in vitro drug release investigation indicated that the liberation of paclitaxel from the nanocomposites was contingent upon both the polymer matrix and the pH of the drug release medium. Furthermore, an in vitro drug release analysis indicated that optimal release depends on the alkaline pH environment [149].

Table 1 showcases a selection of chitosan-based nanocomposites developed to deliver anti-cancer drugs.

### 5.2. Alginate-Based Nanobiocomposites

One kind of bio-polysaccharide found in the ocean is called alginates. Brown algae, such as *Ascophyllum nodosum*, *Laminaria digitata*, *Macrocystis pyrifera*, and *Laminaria hyperborea*, as well as bacteria, contain anionic linear natural polysaccharide groups called alginates [152,153]. Chemically, alginates consist of the linear blocks of (1→4)-linkage between D-mannuronic acid(M) and *α*-L-guluronic acid(G) monomers. These linear blocks are anionic copolymers comprised of α-L-guluronic acid (G unit) as well as *β*-D-mannuronic acid (M unit) arranged in an irregular arrangement of various proportions of GG, MG, and MM units, having 1, 4-glycosidic linkages among these [120] (Figure 11).

The substance in question exhibits biocompatibility, biodegradability, and non-antigenicity. Alginates exhibit the ability to undergo gelation in aqueous solutions. Sodium alginate, the sodium salt derivative of alginic acid, can generate substantial viscous solutions when placed in an aqueous environment [154]. Sodium alginate exhibits a noteworthy characteristic in aqueous environments, wherein it undergoes ionic gelation when exposed to diverse divalent and trivalent metal cations, including but not limited to Ca^2+^, Zn^2+^, Ba^2+^, and Al^3+^ [155]. Alginates have been natural biopolymeric excipients or raw materials in various biomedical fields, including drug delivery, over the past few decades. Various biopolymers and bio-inorganics recently produced nanocomposites based on alginate, enhancing the efficacy of anti-cancer drugs [154].

The study conducted by Illiescu et al. (2014) investigated the utilization of an alginate-montmorillonite (MMT) nanocomposite system as a means of achieving sustained release of irinotecan. Irinotecan is a semi-synthetic derivative of camptothecin, an alkaloid found in nature. Topoisomerase-I inhibition is a common approach in treating various types of cancer, including but not limited to lung, colon, ovarian, rectal, glioma, and malignant tumors. After the integration of MMT and irinotecan, the composite was combined with alginate to create nanocomposite beads of alginate-MMT. The beads were then loaded with irinotecan using the process of ionic gelation. The drying method regulates the size and shape of alginate-MMT nanocomposite beads loaded with irinotecan. The size of the air-dried alginate-MMT nanocomposites containing irinotecan was comparatively smaller than that of the freeze-dried nanocomposite. The nanocomposite comprising MMT and alginate demonstrated a synergistic impact on the extended-release of irinotecan, as observed over a considerable duration [156].

Iliescu et al. (2011) carried out the synthesis of MMT-alginate nanocomposite beads incorporating carboplatin. Carboplatin is a platinum-derived antineoplastic medication used to manage various cancer forms. The synthesized carboplatin-MMT hybrid (drug: clay = 40:60) consists of an aqueous carboplatin solution and the swelled form of MMT. The study involved the development of carboplatin-MMT-alginate nanocomposite beads with a carboplatin-MMT hybrid to-alginate ratio of 15:85. The process involved the gradual incorporation of 0.175 g of carboplatin-MMT hybrid powder into 2% *w*/*v* aqueous solutions of sodium alginate, by applying magnetic stirring at a fixed speed of 600 rpm and a temperature of 60 °C for 4 h to ensure the homogeneity of the resulting solution. The ionic gelation process aids in the synthesis of MMT-alginate nanocomposite beads. [156].

Azhar and Olad (2014) formulated a nanocomposite system consisting of alginate, chitosan, and MMT for the controlled release of 5-fluorouracil. The chemotherapeutic agent 5-fluorouracil, is utilized in the treatment of cancer. In vitro investigations were conducted to examine the impact of the pH levels of the release media and MMT concentrations on the release of 5-fluorouracil. The nanocomposite systems of alginate-chitosan-MMT containing 30% wt MMT and 5-fluorouracil exhibited a protracted drug release profile under pH 7.4 conditions. Within 8 hr, the nanocomposites released 50% of the drug. The present study determined that the nanocomposite systems formulated exhibited a well-fitted 5-fluorouracil releasing profile following the Korsmeyer–Peppas kinetic modeling. This finding suggests that diffusion controlled the release mechanism [157]. Table 2 presents a collection of alginate-based nanocomposites developed to deliver anti-cancer drugs.

### 5.3. Cellulose-Based Nanobiocomposites

Cellulose is a polysaccharide that consists of a linear chain of considerable length, ranging from several hundred to more than 10,000 β-(1-4) connected D-glucose units (Figure 12). Cellulose is a readily available and renewable natural resource in its raw form [160,161]. Cellulose is a ubiquitous organic compound found in many living organisms and obtained from hemp, cotton, linen, and wood sources. The substance in question features a *β* (1, 4) glycosidic linkage between D-(+)-*β*-glucose molecules, resulting in the formation of a polymer of anhydrous-*β*-glucose [59]. Cellulose is a linear, unbranched polysaccharide chain, and many parallel cellulose molecules combine to produce crystalline microfibrils [162]. The mechanically robust and enzymatically resistant crystalline microfibrils (aligned with each other) structured the plant’s and bacterial cell walls [163].

The high molecular weight and intra- and inter-molecular hydrogen bonding of cellulose are responsible for its chemical stability, mechanical strength, biocompatibility, and biodegradability. The utilization of cellulose in tissue engineering presents challenges due to its chemical properties, which render it resistant to dissolution in commonly employed solvents [122]. Cellulose derivatives, namely carboxymethyl cellulose (CMC) and bacterial cellulose, in fabricating scaffolds for tissue engineering purposes resolved the issue, as mentioned earlier [164]. *Acetobacter xylinum* cellulose was isolated using the biosynthetic process. Various nanocomposites derived from cellulose were recently integrated with other biopolymers and bio-inorganics to enhance their efficacy in delivering anti-cancer drugs [162].

In 2015, Madusanka and colleagues produced a new type of nanocomposite by activating curcumin with carboxymethyl cellulose–montmorillonite. Utilization of carboxymethyl cellulose, a biopolymer with super-absorbent properties, was employed as an emulsifying agent for curcumin. The incorporation of montmorillonite, a nano clay material, was utilized to reinforce a biopolymeric matrix of carboxy methyl cellulose. The study revealed a noteworthy enhancement in the water solubility of curcumin when incorporated into the nanocomposite, as compared to its pure form. The study measured the release of curcumin in vitro from a nanocomposite consisting of carboxy methyl cellulose–montmorillonite. The results indicated that 60% of curcumin was released within 2 hr in distilled water with a pH of 5.4 [165].

In 2017, Rasoulzadeh and Namazi created hydrogel beads using carboxymethyl cellulose and graphene oxide bio-nanocomposite. The purpose of this development was to enable the controlled release of DOX. They employed carboxy methyl cellulose as the pH-responsive polymer. The nanocomposite can regulate the discharge of the enclosed medication and carboxymethyl cellulose, which functions as a pH-sensitive polymer and releases the medication at the appropriate physiological pH. They conducted an in vitro cytotoxicity assessment of a formulation on human colon cancer cells (SW480). The DOX-loaded carboxymethyl cellulose/graphene oxide bio-nanocomposite hydrogel beads exhibited promising potential for selectively inducing apoptosis in cancer cells in vitro [166].

Sun et al. (2019) formulated pH-responsive zinc oxide/carboxy methyl cellulose/chitosan nanocomposite hydrogel beads containing 5-fluorouracil for the targeted treatment of colon cancer. The study aimed to develop a colon-specific delivery system for the drug. The process of creating nanocomposite hydrogel beads containing 5-fluorouracil involved the integration of zinc oxide nanoparticles into carboxymethyl cellulose beads, followed by the application of a chitosan-based coating through a self-assembly technique to produce core-shell polyelectrolyte complexes. Analyzing the in vitro swelling and drug release profiles under simulated gastrointestinal conditions allowed researchers to assess the pH-responsiveness of the developed nanocomposite beads. The findings suggest that the nanocomposite hydrogel beads composed of zinc oxide, carboxymethyl cellulose, and chitosan exhibited pH-responsiveness in delivering 5-fluorouracil. The pH-responsive nanocomposite hydrogel beads of 5-fluorouracil, consisting of zinc oxide, carboxymethyl cellulose, and chitosan, possessed biodegradation capability [78].

To accurately regulate the drug release utilizing the CMC/MOF-5/GO bio-nanocomposite, graphene oxide (GO) was modified to increase its surface charge, solubility, and drug loading capacity. The DOX chemotherapeutic agent was encapsulated within the CMC/MOF-5/GO carrier, enhancing anti-cancer properties. The release rate of DOX exhibited a significant increase when exposed to the acidic pH of 5 in the tumor microenvironment, as opposed to the release rate observed at the physiological pH of 7.4. The cytotoxicity test results indicated that the DOX@CMC/MOF-5/GO bio-nanocomposite exhibited a significant level of cytotoxicity toward K562 tumor cells. This bio-nanocomposite may be a promising candidate for the delivery of anti-cancer drugs [167].

Table 3 showcases a selection of newly developed nanocomposites based on cellulose to deliver anti-cancer drugs.

### 5.4. Starch-Based Nanobiocomposites

Starches, derived from plants, are biodegradable polymers that exhibit exceptional biocompatibility. Starches are macromolecular biopolymers characterized by high molecular weight and comprise two prominent structural copolymers, i.e., amylose and amylopectin. Starches possess various biomedical applications, such as wound dressing applications, tissue engineering, and drug delivery. Various potential limitations of starches, such as reduced processability, decreased moisture resistance, and limited stability in acidic environments, are being addressed through the modification of starches. Synthesized starch-based nano-composites involve the reinforcement of other biopolymers and bio-inorganics, delivering anti-cancer drugs.

Gholamali and Yadollahi (2020) fabricated hydrogel beads composed of carboxymethyl cellulose, starch, and zinc oxide nanocomposites in their study. The purpose of this was to achieve a controlled release of doxorubicin. The synthesized nanocomposite hydrogel beads utilize ferric chloride to cross-link starch, carboxymethyl cellulose, and zinc oxide nanoparticles. The study demonstrated that the hydrogels’ drug release and swelling characteristics were influenced by the concentration of carboxymethyl cellulose, pH levels, and the presence of zinc oxide nanoparticles, as observed through in vitro testing. The study observed extended and regulated drug release patterns in carboxymethyl cellulose/starch beads containing zinc oxide nanoparticles, which exhibited increased drug release duration with an increase in the concentration of zinc oxide nanoparticles. The confirmation of the samples’ cytotoxicity was conducted through the utilization of human colon cancer cells (SW480) [172].

In a separate study, Subramanian et al. (2014) developed nanocomposite particles containing bisdemethoxy curcumin analog through the use of various ratios of chitosan and starch (3:1, 1:1, and 1:3) utilizing the ionic gelation technique. The formulation containing a ratio of 3:1 of bisdemethoxy curcumin analog to chitosan exhibited high entrapment efficiency and drug loading capacity. SEM analysis revealed that the morphology of the nanocomposite particles loaded with bisdemethoxy curcumin analog and composed of chitosan–starch was consistently spherical and regular. The drug release patterns of the bisdemethoxy curcumin analog loaded onto chitosan–starch nanocomposite particles were observed in vitro, indicating a controlled diffusion process that resulted in a slow and sustained drug release. The in vitro drug release profile of the BDMCA-CS nanocomposite particles showed a very slow and sustained diffusion-controlled drug release. The cancer cells targeting ability of the BDMCA-CS nanocomposite particles exhibited more potent cytotoxicity, including cell death against the MCF-7 cell line than on the VERO cell line [173].

In 2016, Raj and Prabha created nanocomposites using cassava starch acetate, polyethylene glycol (PEG), and gelatin to deliver cisplatin. The present study uses a straightforward nanoprecipitation technique to describe the development of nanocomposites comprising cassava starch acetate-PEG/gelatin and cisplatin. The Zetasizer instrument measures the particle sizes of the cisplatin-entrapped cassava starch acetate, cisplatin-entrapped cassava starch acetate-PEG, and cisplatin-entrapped cassava starch acetate-PEG/gelatin polymer composites. The measured particle sizes ranged from 140 to 350 nm. The findings of this study indicate that the utilization of cross-linked cassava starch acetate-PEG/gelatin nanocomposites can potentially serve as a polymeric vehicle for the controlled administration of cisplatin [174].

Saikia et al. (2017) produced a nanocomposite of aminated starch, zinc oxide, and iron oxide coated and tagged with folic acid. The purpose of this nanocomposite was to serve as a targeted delivery system for curcumin. The present investigation involved the amination of starch as a primary step, followed by the coating of the resulting aminated starch solution and the pre-swelled and sonicated zinc oxide solution onto the iron oxide core, and later incorporated curcumin into the nanocomposite’s aqueous dispersion. The investigation into the viability of human lymphocytes concerning the nanocomposite indicated that it is biocompatible. Viability and uptake assays were conducted on two distinct cancer cell lines, namely MCF7 and HepG2 cells, to assess the efficacy of folic acid-targeted nano-composites. The results obtained from both cell lines were significant [175].

Increased methotrexate (MTX) oral bioavailability with nanocomposite films comprising starch and PEC. Enhanced puncture resistance was quantified; however, the barrier characteristics demonstrated reduced permeability to water vapor. The ex vivo bio-adhesion analysis results demonstrated a robust interaction between the nanocomposite films and the mucosal lining of the porcine gastrointestinal tract. The in vitro drug release test results demonstrated that the films exhibited an improved drug dissolution profile, releasing approximately 80% MTX within 150 min. This finding highlights the potential of such films as carriers for drugs with poor solubility, which could ultimately enhance the oral bioavailability of such drugs [176].

Table 4 presents a collection of starch-based nanocomposites developed to deliver anti-cancer drugs.

### 5.5. Xanthan Gum-Based Nanobiocomposites

Xanthan gum is a biopolymer obtained from the fermentation of extracellular polysaccharides in the presence of sugars such as glucose and sucrose, facilitated by the bacterium *Xanthomonas campestris*. Xanthan gum’s primary structure comprises pentasaccharide subunits that consist of D-mannosyl, D-glucosyl, and D-glucuronic acid residues. These residues are present in ratios of 2:2:1 and associated with varying proportions of *O*-acetyl and pyruvyl residues (Figure 13) [35,180].

Xanthan gum possesses a linear structure devoid of branching and comprises D-glucose (1-4) linkage. This structure bears a striking resemblance to the backbone of cellulose and is capable of undergoing conformational transitions upon thermal induction [35]. This product demonstrates remarkable water solubility and displays favorable biocompatibility. The broad spectrum of pH values, ionic concentrations, and temperature ranges enhances the efficacy of the substance. The range of its potential applications encompasses a wide variety of fields, including both the pharmaceutical and food sectors [181]. It develops diverse drug delivery modalities, particularly emphasizing oral, nasal, brain, buccal, and other delivery systems. In recent times, there has been a development in the creation of starch-based nanocomposites to deliver anti-cancer drugs [182,183].

### 5.6. Pectin-Based Nanobiocomposites

Pectins are a complex polysaccharide naturally present in plant cell walls extracted from various sources, such as citrus peels, sugar beet roots, and apple pomaces. They are water-soluble, non-starch, and non-toxic, making them an inexpensive and natural option for use as thickening agents, food additives, and gelling agents in industrial applications [184]. It has a linear polymer structure consisting predominantly of alpha (1, 4) glycosidic linkages of D-galacturonic acid residues. The galacturonic acid backbone features sporadic rhamnose moieties that impede the formation of helical structures within the chain. Additionally, α-L-rhamnopyranose is present through α-(1-2) linkage, with a molecular size ranging from several hundred to roughly one thousand building blocks (Figure 14) [185,186].

Methoxy groups predominantly esterify the pectin backbone’s galacturonic acid residue’s carboxylic acid groups in pectin’s natural product. Introducing a significant quantity of sucrose (>50%) in an acidic environment enhances the gelling property of high methoxy pectins. Low methoxy pectins can create gel structures through ionotropic gelation with divalent cations, such as Ca^2+^ and Zn^2+^. These cations can be potential vehicles in drug-delivery applications [187,188]. The interactions between the carboxyl groups of the low methoxy pectin backbone and divalent ionotropic cross-linking cations induce the formation of the “Egg-Box” structure. However, it exhibits slight variations from the “Egg-Box” model initially established for ionotropic gelation of sodium alginate [184,189,190]. For several years, pectin has been a pharmaceutical excipient used in developing various dosage forms, including but not limited to beads, tablets, nanoparticles, microparticles, hydrogels, gels, films, scaffolds, and patches [191]. Developed pectin-based systems target and deliver drugs to the colon. In recent times, there has been a fabrication of several nanocomposites based on pectin to deliver anti-cancer drugs [192,193].

Wang et al. (2020) have devised a method for administering 5-fluorouracil orally, explicitly targeting the colon. This delivery system utilizes pectin/modified nanocarbon sphere-based nanocomposite gel films. The 3-Amino propyl tri-ethoxy silane-modified nanocarbon sphere-based pectin-Ca^2+^ film enhanced the controlled release properties of the pectin-based system. The results obtained from the Fourier-transform infrared (FT-IR) measurements demonstrated the efficacious alteration of the nanocarbon sphere through the process of silylation, as well as the electrostatic interplay between the pectin molecules and the modified nanocarbon sphere within the nanocomposite film. The field emission scanning electron microscope (FE-SEM) visualizes the pore structure from combined modified nanocarbon spheres and pectin. The range of encapsulation efficiency values observed was between 30.1% and 52.6%. The composite film-based oral colon-specific drug-delivery systems exhibited superior encapsulation efficiency compared to the single pectin-Ca^2+^-based oral colon-specific drug-delivery system. The drug release studies’ findings indicate that most oral colon-specific drug-delivery systems derived from composite films exhibited superior release characteristics to single pectin-Ca^2+^-based oral colon-specific drug-delivery systems. The results indicated that the optimized sample exhibited the most favorable release performance, as evidenced by the cumulative release rates of 22.77%, 32.17%, and 63.89% in simulated small intestinal fluid, gastric fluid, and colon fluid, respectively. The cytotoxicity assay results suggest that the composite carriers exhibit favorable biocompatibility [194].

### 5.7. Guar Gum-Based Nanobiocomposites

Guar gum is a naturally occurring, non-ionic polysaccharide gum derived from the seeds of *Cyamopsis tetragonoloba*, a plant species belonging to the Leguminosae family. Its distinguishing features include its ability to dissolve in water [195]. Guar gum possesses a structure characterized by linear polymeric chains consisting of (1→4)-β-D-mannopyranosyl units and associated branching wherein α-D-galactopyranosyl units are linked by (1→6) linkages (Figure 15) [196].

The residues of galactomannan exhibit significant fermentability by the microbiota in the gastrointestinal tract (GIT) [197,198,199]. Guar gum exhibits advantageous characteristics such as biodegradability and biocompatibility [200,201]. The substance exhibits various characteristics, including but not limited to its mechanical strength, physicochemical stability, and bioavailability. Guar gum is a suspending, thickening, emulsifier, and stabilizing agent [202,203,204]. It enhances diverse hydrophilic matrices in developing controlled drug-release oral delivery vehicles [205,206,207,208,209]. Its efficient gelling properties and enzymatic degradation in colonic fluids make it a suitable candidate for drug delivery [210]. Guar gum’s excellent gelling network structure decelerates drug release patterns in different dosage forms. Chemical and physical functionalizations of guar gum regulate its swelling properties in diverse buffer solutions. This approach develops orally administered drug-delivery systems with colon-targeting capabilities. Several nanocomposites based on guar gum deliver anti-cancer drugs [207,211].

### 5.8. Hyaluronic Acid-Based Nanobiocomposites

Hyaluronic acid (or hyaluronate) is a non-sulfated glycosaminoglycan, anionic, comprising repeated units of “D-glucuronic acid-*β*-1, 3-N-acetyl-D-glucosamine-*β*-1, 4” units (Figure 16) [120].

The substance in question is present within the extracellular matrix of all connective tissues throughout the human body. It possesses some notable characteristics, including exceptional viscoelasticity, biocompatibility, solubility in water, and a lack of immunogenicity [212]. However, enzymatic activity causes its degradation. Synthesized nanocomposites based on hyaluronic acid deliver anti-cancer drugs. The authors of the study developed a chemotherapeutic agent named high-penetrating cationic liposomes (HPCID), which is composed of a nanocomposite consisting of hyaluronic acid (HA), carboxyl terminated dendrimer, fluorochrome indocyanine green, and DOX. The HPCID demonstrated effective targeting of metastatic cancer cells and exhibited an enhanced therapeutic effect. The sonochemotherapy method utilized DDS to CD44 overexpressed metastatic 4T1 breast cancer cells in vivo and in vitro. Incorporating HA into the drug-delivery system (DDS) significantly improved the internalization of HPCID by the cells.

Additionally, the HA shell was degraded by the abundant hyaluronidase in 4T1 cells, leading to a drug release that was responsive to the enzymatic activity. The HPCID induced significant cellular apoptosis by generating high levels of reactive oxidant species through ultrasound radiation and chemotherapy. In addition, the administration of HPCID to mice bearing 4T1 xenografts, in conjunction with ultrasonic radiation, significantly inhibited tumor growth and pulmonary metastasis without any observed systemic toxicity. Therefore, the utilization of HPCID-mediated sono-dynamic therapy proved to be a potent strategy in combating the metastasis and progression of breast cancer [213]. Rao et al. (2014) developed a polymeric nanocomposite hydrogel by combining halloysite-sodium sodium hyaluronate/poly (hydroxyethyl methacrylate) to deliver 5-fluorouracil for the treatment of colon cancer. One notable benefit of this nanocomposite is its capacity to encapsulate drugs within the hydrogel and the halloysite nanotube shell. The process of incorporating 5-fluorouracil into halloysite nanotubes and hydrogel involved subjecting the nanocomposite to a vacuum, subsequently followed by the application of tension and release of the vacuum. The drug-delivery system under consideration exhibits pH sensitivity, whereby it undergoes rapid drug release upon encountering the pH conditions of the colon. The 5-fluorouracil utilizes this property for oral administration [214].

### 5.9. Gelatin-Based Nanobiocomposites

Gelatin is also another naturally occurring biodegradable and multi-functional biopolymer [215]. Gelatin is a thermoreversible form. The substance in question is a composite of proteins and peptides that have undergone partial hydrolysis of collagen. This collagen is sourced from specific animals’ bones and skin remnants [215,216]. The polymeric chain of gelatin contains a significant amount of hydroxyproline, proline, and glycine, rendering it a substantial reservoir of these amino acids (Figure 17).

There are two classifications of gelatin: type A and type B. Hydrolysis types are classified based on their respective mechanisms, with acid hydrolysis as type A and base hydrolysis as type B. The sol–gel transformation property of gelatin is the thermoreversible characteristic. The primary application of this substance is prevalent in the food industry, where it serves as a thickener and an emulsifier in both hard and soft gelatin capsules [215]. Gelatin has been utilized in drug-delivery systems by creating hydrogels due to its swell ability. These inherent properties facilitate the preparation of films, microencapsules, and nanoparticles [217]. These applications are relevant to a range of drug delivery methods, such as those employed in anti-cancer, ocular, and pulmonary contexts, as well as enzyme, protein, and gene delivery. Several synthesized nanocomposites comprising hyaluronic acid-delivering anti-cancer drugs [217,218,219].

### 5.10. Albumin-Based Nanobiocomposites

Albumin is a significant protein polymer that occurs naturally and is recognized as the most abundant protein in the bloodstream [220]. The structure in question functions as a reservoir and is additionally involved in transporting various substances, including hormones, nutrients, toxins, and metals. The production of albumin primarily occurs in liver hepatocytes [221]. Three distinct types of albumin are currently available: ovalbumin, bovine serum albumin, and human serum albumin [220]. The monomeric phosphoglycoprotein known as ovalbumin has applications in food and drug delivery due to its affordability, accessibility, ability to stabilize emulsions, and responsiveness to changes in pH and temperature [221]. Due to its capacity to interact with ligands, bovine serum albumin is extensively employed in the delivery of drugs [222,223]. Human serum albumin, the most abundant hydrophilic plasma protein, is a significant constituent in various drug-delivery systems due to its biodegradable and inert properties. In recent times, albumins have created nanoparticles employed in various drug-delivery applications, including the delivery of anti-cancer drugs [221].

### 5.11. Collagen-Based Nanobiocomposites

Collagen is a naturally occurring polymer and one of the living body’s most abundant structural proteins. It is a principal component of the extracellular matrix (ECM) [224]. This phenomenon typically manifests in the connective tissues of the tendon, cartilage, and bone [224,225,226]. Collagen is a protein-based biomaterial capable of undergoing biodegradation [212]. Collagen is a popular biomaterial in biomedical applications due to its inherent biocompatibility, characterized by low antigenicity, non-inflammatory properties, and minimal cytotoxic responses. Its versatility has led to its use in various applications, such as tissue engineering for soft and hard tissues, wound healing, and drug delivery [227,228].

Nevertheless, it exhibits certain constraints, such as diminished mechanical strength and inferior elastic properties. Various strategies address the constraints associated with collagen as a biomaterial, including using collagen-based composites through preparation or synthesis. Recently, there has been a proliferation of collagen-based nanocomposites which deliver anti-cancer drugs [212].

### 5.12. Miscellaneous

Nanotechnology is a promising area in the twenty-first century, requiring a systematic restructuring of novel applications in various fields such as pharmacy, energy storage, inorganic and organic materials, semiconductors, and biotechnology [229]. Nanomedicines have demonstrated significant potential compared to conventional therapeutic agents for cancer treatment. Nanomaterials, including but not limited to gold nanoparticles, MoS_2_, iron oxide, MnO_2_, and carbon nanotubes, have been extensively researched and developed for various applications in cancer detection, imaging diagnosis, and treatment [230]. Polyesters derived from camphoric acid and group IV B metallocene dichlorides exhibit significant efficacy in suppressing various cancer cell lines, including two distinct pancreatic cell lines. This group of compounds can potentially serve as a novel class of anti-cancer medications [231].

Recently, nanotechnology has garnered heightened interest due to its efficacy in diagnosing and treating select cancerous tumors. Nanocarriers are substitutes for traditional antitumor drug-delivery systems owing to their non-specificity, absence of adverse effects, rapid release, and minimal harm to healthy cells. Nanocarriers enhance antitumor drugs’ therapeutic efficacy by facilitating preferential accumulation at the intended site. A restricted quantity of nanocarriers has received clinical approval for their intended functions at specific sites [232]. This analysis involved the preparation of polyurethane (PU)–DOX nanoparticles in an aqueous medium. The process utilizes electrostatic interactions between amphiphilic PU and carboxyl pending groups (PU–COOH and DOX HCl). The experimental results indicate that PU-DOX nanoparticles exhibit superior cellular internalization and more significant inhibition of breast cancer (MCF-7) cell proliferation compared to pure DOX [233]. The inorganic components and biotin contained in the macromolecular chain are responsible for the pH sensitivity and tumor-targeting properties of heparin–biotin/heparin/CaCO_3_/Ca phosphate/deoxyribonucleic acid (DNA)/DOX nanoparticles [234]. Amphiphilic polymers composed of PEG and bis-pyrene with a disulfide (SS) bond as a reduction linker produced a novel nanoparticle system. This system has demonstrated the ability to increase the reactive oxygen level in cancer cells upon exposure to UV irradiation, ultimately resulting in apoptosis. The highest degree of release correlates to an endosome pH of 5.5 [235].

Another study reported the antitumor targeting and therapeutic effects of polyaminoamide dendrimers. The dendrimers were partially neutralized on their surfaces utilizing covalent bonding with 2,3-dimethylmaleic anhydride. Subsequently, DOX was encapsulated in these nano aggregates to evaluate their efficacy in antitumor targeting and therapy. These nano aggregates can achieve significant deformation without external stimuli, thereby exhibiting potential for employment in cancer therapy [236]. The fabricated novel nanofiber comprises of poly(vinyl pyrrolidone) shell and a hyperbranched poly(ethylene adipate) core, which encapsulates the artemisinin (ART) drug. The MTT assay, utilizing [3-(4,5 dimethyl-2 thiazolyl) (2,5 diphenyl-2H-tetrazolium bromide)], demonstrated that the polymer nanocomposite exhibited a reduction in viability of prostatic cancer cells [237,238,239].

## 6. Challenges and Future Perspectives of Natural Polymeric Nanobiocomposites for the Delivery of Anti-Cancer Drugs

The utilization of diverse nanomaterials possessing desirable properties, coupled with advancements in drug delivery, has presented significant obstacles in cancer treatment. The advancements in drug delivery over the past few years have led to the expectation that nanomaterials will revolutionize the entire healthcare system. The development of effective cancer nanotherapeutics presents a significant obstacle, with only a limited number of nanoformulations progressing to clinical trial phases. The physicochemical characteristics of nanomaterials play a crucial role in determining their biocompatibility and toxicity in biological systems. The cautious production of nanomaterials for drug delivery is necessary to prevent the potential toxicity of nanocarriers to the human body. Using nanocarriers in cancer treatment may result in undue toxicity due to aggressive interactions with biological entities. Several research studies have revealed the adverse characteristics of nanocarriers due to their toxic nature. The field of nanotoxicology, which falls under the umbrella of nanomedicine, has emerged as a crucial area of investigation. It offers a tangible methodology for assessing the potential toxicity of nanoparticles. A significant and persistent issue is the absence of a correlation between drug release profiles in vitro and in vivo. Furthermore, producing nanomedicine products for commercial purposes is a significant obstacle due to the difficulty of achieving large-scale manufacturing [119].

An additional crucial matter pertains to the regulatory clearance of nanotherapeutics, given the absence of established directives by the FDA for nanomaterial-based commodities. The present criteria use guidelines that pertain to bulk materials. Regulatory decisions regarding nano-formulated drugs rely on individual evaluations of benefits and risks, resulting in a protracted assessment process that delays commercialization. Moreover, the challenges in obtaining approval will likely escalate due to the advancement of multifunctional nanoplatforms. Therefore, to address the challenges associated with using nanomaterial-based therapeutic agents for cancer treatment, it is imperative to implement design and development strategies before their application in the medical field, to enhance treatment efficacy and improve human health outcomes. Comprehending the intricacies inherent in the physiology of cancer cells and the microenvironment of tumors, in conjunction with the pharmacokinetics of drugs and carriers, is imperative in creating efficacious novel cancer treatments. Furthermore, individualized examinations are necessary to utilize the vast capabilities of cancer nanotherapeutics fully. There is an urgent need for a comprehensive set of guidelines to facilitate the evaluation and approval of cancer nanotherapeutics for regulatory purposes [119].

Significant challenges and limitations in delivering and developing a nanocarrier can be summarized as follows:(a)Nanomaterials’ physicochemical characteristics are fundamental in deciding their biocompatibility and toxicity in biological systems;(b)The vigilant fabrication of nanomaterials for drug delivery is compulsory to check the latent toxicity of nanocarriers to the human body;(c)The lack of a correlation between in vitro drug release profiles with in vivo data is a noteworthy and unrelenting problem;(d)The regulatory judgments concerning nano-formulated drugs are contingent on individual evaluations of benefits and risks, resulting in a protracted assessment process that delays commercialization. Moreover, the challenges in obtaining approval will likely rise due to the development of multifunctional nanoplatforms.

## 7. Conclusions

One of the primary challenges in cancer diagnosis and treatment is minimizing the need for surgical intervention and mitigating the potential for adverse side effects. Utilizing nanocomposites aims to enhance drug-delivery systems and diagnostic markers’ sensitivity. Recently, there has been a growing interest in utilizing nanocomposite materials to deliver anti-cancer drugs, particularly in cancer treatment. The field of drug therapy for cancer treatment has shown an apparent inclination toward utilizing magnetic nanoparticles to achieve targeted drug delivery to tumor cells. This approach is practical, although integrating imaging systems could further enhance the precision of drug delivery to the tumor cells. Because of advances in nanotechnology that allow for the development of complex therapeutic composites, it is possible to improve the effectiveness of cancer treatment; however, multicomponent nanosystems necessitate a thorough understanding of the processes that occur during their preparation to prevent some aggregation. Anti-cancer natural nano biocomposites exhibit better anti-cancer activity with minimum toxic effects than pure compounds. In addition, natural nano biocomposites are highly biocompatible with normal cells.

## Figures and Tables

**Figure 1 pharmaceutics-15-02064-f001:**
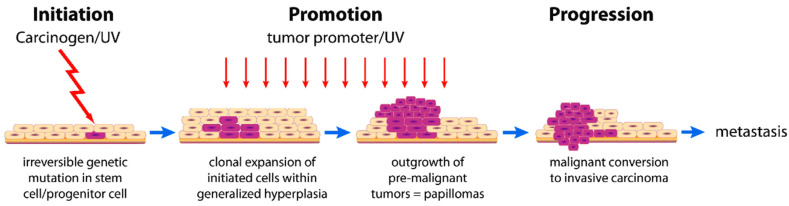
Mechanisms of different phases of cancer cells: initiation, promotion, and progression (Reused from Rundhaug and Fischer, 2010, distributed under the usage, distribution, and reproduction are all allowed without restriction in any format as per the Creative Commons Attribution 4.0 International License. http://creativecommons.org/licenses/by/4.0/ (accessed on 3 May 2023)) [53].

**Figure 2 pharmaceutics-15-02064-f002:**
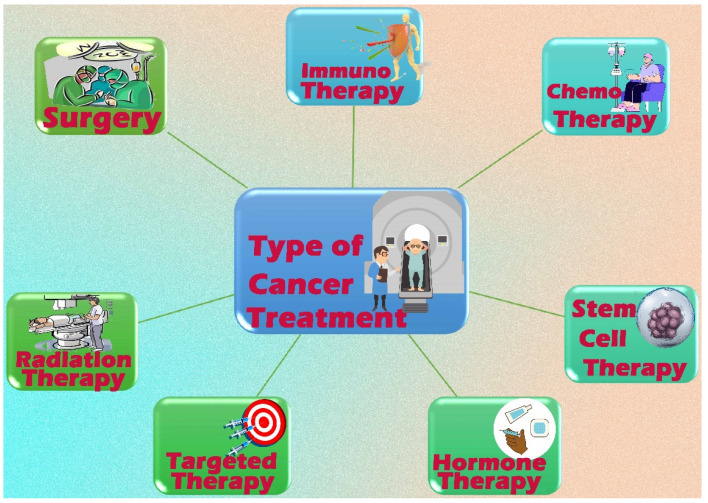
Representation of cancer therapy utilizing systemic treatments (chemotherapy, hormone treatment, targeted therapy, and immunotherapy) and local therapies (surgery, radiotherapy, and phototherapy).

**Figure 3 pharmaceutics-15-02064-f003:**
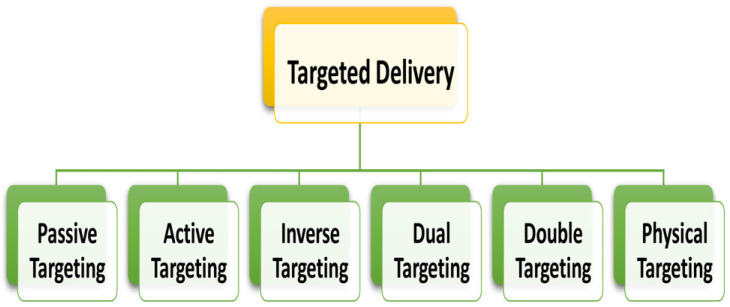
Strategies of targeted drug-delivery system.

**Figure 4 pharmaceutics-15-02064-f004:**
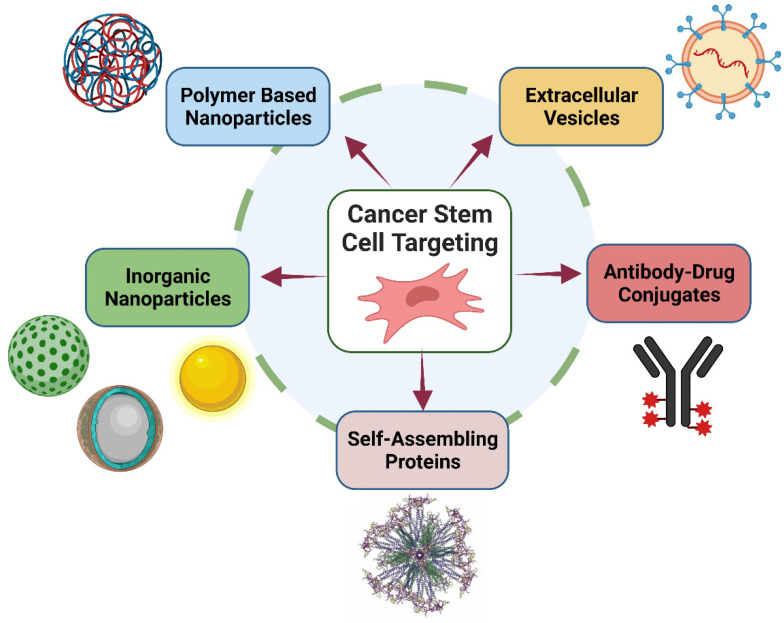
Different approaches to delivering anti-cancer drugs to the tumor sites via nanoparticles (reused from Ertas et al., 2021, distributed under Creative Commons Attribution 4.0 International License, which permits unrestricted use, distribution, and reproduction in any medium. http://creativecommons.org/licenses/by/4.0/ (accessed on 3 May 2023)) [77].

**Figure 6 pharmaceutics-15-02064-f006:**
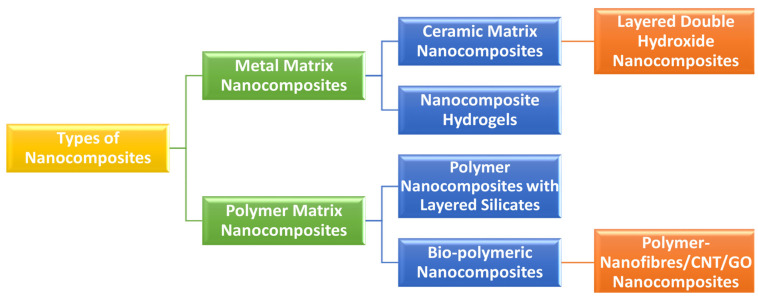
Scheme of various types of nanocomposites.

**Figure 7 pharmaceutics-15-02064-f007:**
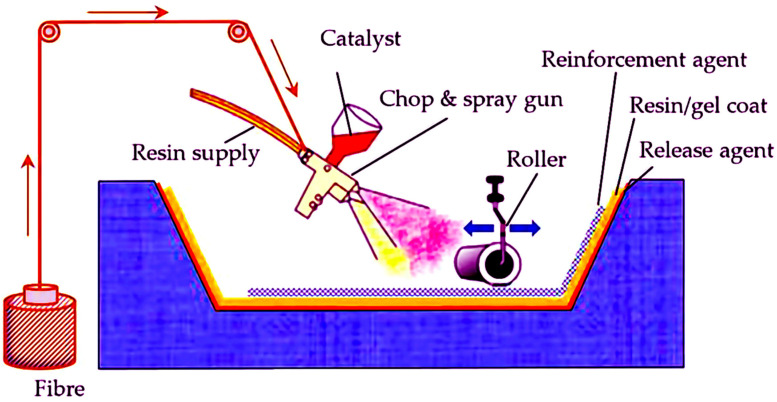
The schematic of the spray lay-up process (reused from Ngo, 2020, distributed under Creative Commons Attribution 3.0 International License, which permits unrestricted use, and redistribution in any medium. http://creativecommons.org/licenses/by/3.0/ (accessed on 3 May 2023)) [111].

**Figure 8 pharmaceutics-15-02064-f008:**
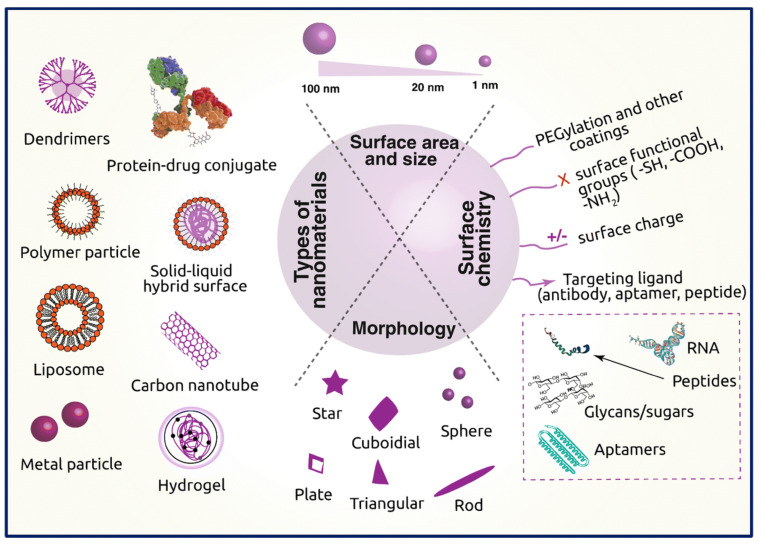
A schematic depiction of various categories of nanomaterials utilized in cancer treatment, highlighting their crucial physical characteristics, as well as the requisite surface chemistry for drug delivery (reused from Navya et al., 2019, distributed under Creative Commons Attribution 4.0 International License, which permits unrestricted use, distribution, and reproduction in any medium. http://creativecommons.org/licenses/by/4.0/ (accessed on 3 May 2023)) [119].

**Figure 9 pharmaceutics-15-02064-f009:**
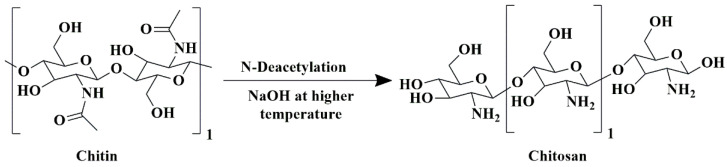
Structure of chitosan.

**Figure 10 pharmaceutics-15-02064-f010:**
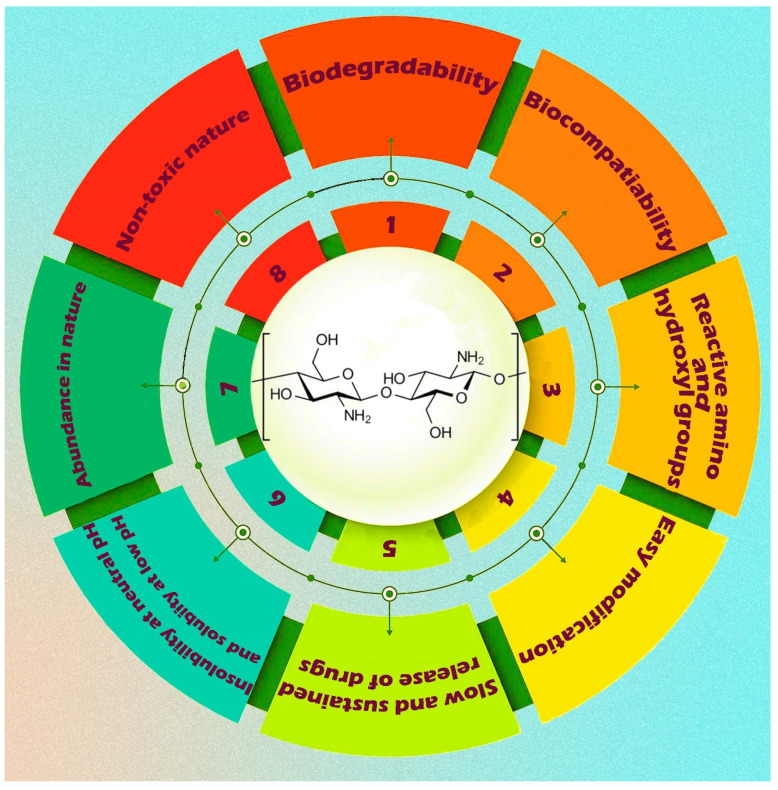
Graphical depiction of a chitosan-based nanocomposite that included diverse kinds of nanofillers.

**Figure 11 pharmaceutics-15-02064-f011:**
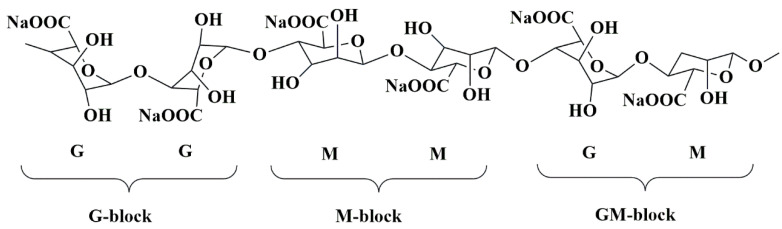
Types of alginate blocs: G = guluronic acid; M = mannuronic acid.

**Figure 12 pharmaceutics-15-02064-f012:**
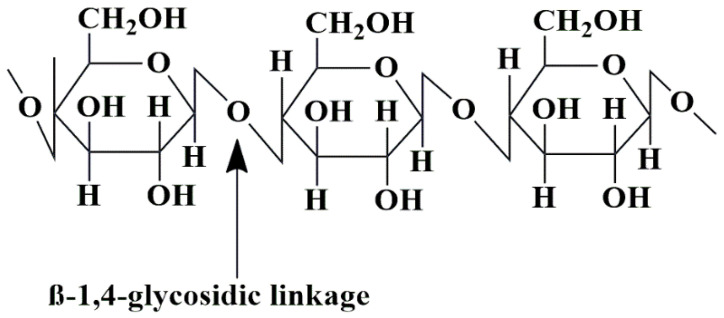
Cellulose is a polysaccharide with a linear, elongated chain consisting of β-(1-4) linkages between D-glucose units.

**Figure 13 pharmaceutics-15-02064-f013:**
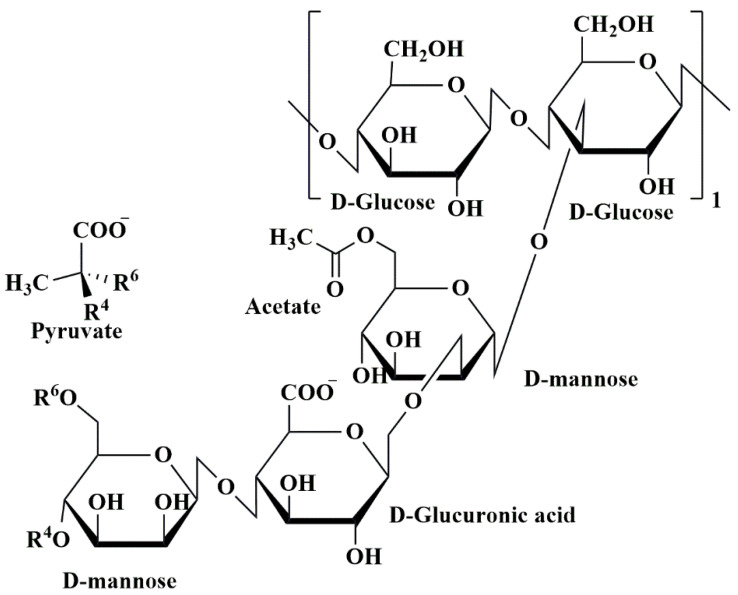
Xanthan gum: D-mannosyl, D-glucosyl, and D-glucuronic acid residues in a ratio of 2:2:1. The gum also contains varying proportions of *O*-acetyl and pyruvyl residues.

**Figure 14 pharmaceutics-15-02064-f014:**
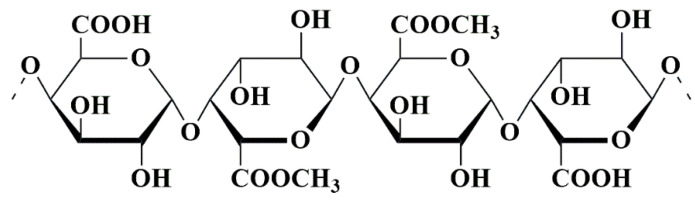
Structure of pectin.

**Figure 15 pharmaceutics-15-02064-f015:**
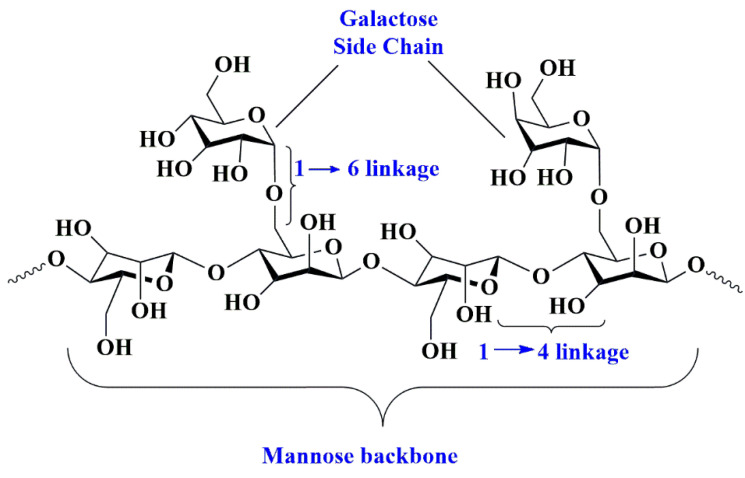
Guar gum has a structural framework comprising linear polymeric chains of (1→4)-β-D-mannopyranosyl units. Additionally, branching is observed in the structure, facilitated by α-D-galactopyranosyl units connected by (1→6) linkages.

**Figure 16 pharmaceutics-15-02064-f016:**
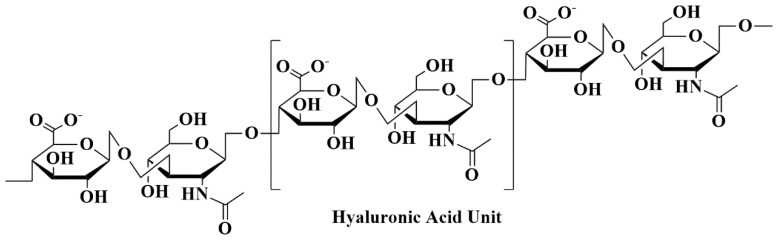
The structure of hyaluronic acid.

**Figure 17 pharmaceutics-15-02064-f017:**
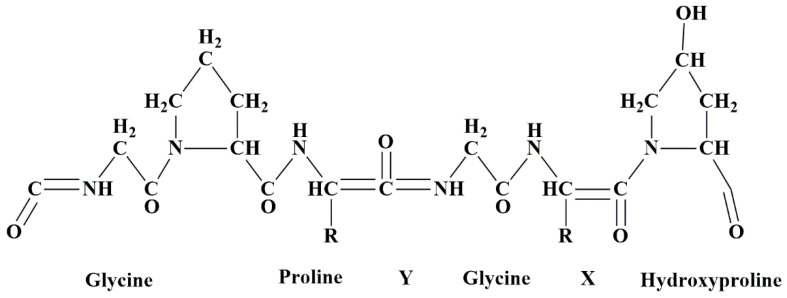
Structure of gelatin.

**Table 1 pharmaceutics-15-02064-t001:** Some recently developed chitosan-based nanocomposites for anti-cancer drug delivery.

Nanobiocomposites for Anti-Cancer Drug Delivery	Drugs	References
Ag_2_S(DOX)@CS nanospheres	Doxorubicin	[150]
Stimuli-responsive hydrogel nanocomposite with magnetic Fe_3_O_4_ nanoparticles	Doxorubicin	[144]
Biomass-derived dialdehyde cellulose cross-linked chitosan-based nanocomposite hydrogel with phytosynthesized zinc oxide nanoparticles (ZnO NPs)	Curcumin	[151]

**Table 2 pharmaceutics-15-02064-t002:** Few alginate-based nanocomposites have been synthesized recently for anti-cancer drug delivery.

Nanobiocomposites for Anti-Cancer Drug Delivery	Drugs	References
CaP nanocomposites with alginate as a polymer template	Chlorogenic acid (CG-NP), caffeic acid (CA-NP), or cisplatin (CP-NP)	[158]
Alginate/clay/imidazolium-based ionic liquid-based nanocomposites	Methotrexate	[159]

**Table 3 pharmaceutics-15-02064-t003:** Few cellulose-based nanocomposites developed recently for anti-cancer drug delivery.

Nanobiocomposites for Anti-Cancer Drug Delivery	Drugs	References
Nanocomposite of carboxymethyl cellulose (CMC) and core-shell nanoparticles of Fe_3_O_4_@SiO_2_	Quercetin	[168]
The cellulose-based nanocomposite of magnetic (Fe_3_O_4_), zinc oxide (ZnO)	Doxorubicin	[169]
Carboxymethyl cellulose/graphene quantum dot nanocomposite	Doxorubicin	[167]
Nanocomposites of Fe_3_O_4_-supported on rice straw cellulose	5-fluorouracil	[170]
ZnO/carboxymethyl cellulose/chitosan bio-nanocomposite	5-fluorouracil	[78]
Cellulose grafted hydrogel doped calcium oxide nanocomposites	Doxorubicin	[171]
Nanocomposite of carboxymethyl cellulose (CMC) and core-shell nanoparticles of Fe_3_O_4_@SiO_2_	Quercetin	[168]

**Table 4 pharmaceutics-15-02064-t004:** Few starch-based nanocomposites developed recently for anti-cancer drug delivery.

Nanobiocomposites for Anti-Cancer Drug Delivery	Drugs	References
Nanocomposite of myco-synthesized copper nanoparticles (CuNPs) and starch	Copper	[177]
Carboxymethyl cassava starch (CMCS)-functionalized magnetic nanoparticles	Doxorubicin hydrochloride	[178]
Starch-based stimuli-responsive magnetite nanohydrogel (MNHG), namely Fe_3_O_4_-g-[poly(N-isopropylacrylamide-co-maleic anhydride)]@strach; Fe_3_O_4_-g-(PNIPAAm-co-PMA)@starch	Doxorubicin	[179]

## Data Availability

The data presented in this study will be provided without restrictions upon communication with the corresponding author.

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
