# Peer review of "Natural Polymeric Nanobiocomposites for Anti-Cancer Drug Delivery Therapeutics: A Recent Update"

_pharmaceutics, 2023, doi:10.3390/pharmaceutics15082064_

Round 1
Reviewer 1 Report
-> Figure 1 and 3 seems less clear and legible, please revise them keeping more clear and readable figures.
-> Please explain in the background why this review is different from other recently reported reviews.
->The section challenges and future perspectives should be discussed with subsections highlighting the main points.
-> What are the authors' thoughts, among the different therapeutics discussed, which one is most suitable or improved for the anticancer drug delivery therapeutics in the context of natural polymeric nano-biocomposites?
->The introduction and discussion section also demand some more input and corrections. Especially, the discussion section has missed many sections regarding inputs from the authors as well as referring to previous literature.
->The grammatical errors and typing errors should be rechecked, therefore revise the manuscript to eradicate all those mistakes.
-> Some in-depth concluding remarks should be added by keeping the limitations and modifications of the reported systems.
Author Response
RESPONSE TO REVIEWERS’ COMMENTS
Manuscript ID: 2401963
Natural Polymeric Nano-biocomposites for Anticancer Drug Delivery Therapeutics: A Recent Update
The authors of this manuscript express their sincere thanks to the Guest Editor and the reviewers for the critical assessment of this work. The authors have acted upon the recommendations of the Guest Editor and the reviewers which have resulted in a significant enhancement in the quality of this manuscript. All modifications incorporated in the manuscript are highlighted in red color font. A “point-by-point” response to each and every comment is outlined below.
Reviewer 1
Comment 1: Figure 1 and 3 seems less clear and legible; please revise them keeping more clear and readable figures.
Response: Figures 1 & 3 were edited and now both the pictures are clear and legible (Page 4 & 7)
Comment 2: Please explain in the background why this review is different from other recently reported reviews.
Response: We have discussed in the introduction section that how this review is different from other reviews (Page no.3, lines 101-114)
Comment 3: The section challenges and future perspectives should be discussed with subsections highlighting the main points.
Response: As suggested by the reviewer we have mentioned point wise (Page no.35, lines 1349-1361)
Comment 4: What are the authors' thoughts, among the different therapeutics discussed, which one is most suitable or improved for the anticancer drug delivery therapeutics in the context of natural polymeric nano-biocomposites?
Response: We have included the suitable drug delivery therapeutics in conclusion section [Page no.35, lines-1368-1370]
Comment 5: The introduction and discussion section also demand some more input and corrections. Especially, the discussion section has missed many sections regarding inputs from the authors as well as referring to previous literature.
Response: We have tried to address all the raised issues related to balanced, comprehensive and critical view of the available literature in the field. We have rectified and corrected as per instructions. Introduction section is based on nanoparticles and its diversity and application, whereas in the 2nd paragraph we only emphasized on various types and its applications of nanocomposites. So, both are connected and relevant with respect to this review article. Further we have included some additional references.
Comment 6: The grammatical errors and typing errors should be rechecked, therefore revise the manuscript to eradicate all those mistakes.
Response: We sincerely apologize for the inadvertent errors. We have extensively edited our manuscript to limit typographic and grammatical errors.
Comment 7: Some in-depth concluding remarks should be added by keeping the limitations and modifications of the reported systems.
Response: We have added some concluding remarks in the conclusion section.
Reviewer 2 Report
Several points below should be addressed before the review could be considered to be published.
1. Line 43-48, these sentences don't make sense.
2. The 2nd paragraph appears unnaturally with little connection to the 1st paragraph. The authors should add more discussion.
3. Section 2 Cancer should be shortened to some degree.
4. The resolution of figure 7 is not enough.
5. The author should draw the structure of all the polysaccharides using the same style, but not showing in the review with different styles.
6. Line 184-186, one recent review (Advanced Materials 33 (18), 2005513) should be included to support such a claim.
Moderate editing of English language
Author Response
2nd Reviewer’s Comments:
Several points below should be addressed before the review could be considered to be published.
Comment 1: Line 43-48, these sentences don't make sense.
Response: We have removed the sentences.
Comment 2: The 2nd paragraph appears unnaturally with little connection to the 1st paragraph. The authors should add more discussion.
Response: Rectified and corrected as per instructions. Introduction section is based on nanoparticles and its diversity and application, whereas in the 2nd paragraph we only emphasized on various types and its applications of nanocomposites. So, both the paragraphs are connected and relevant with respect to this review article.
Comment 3: Section 2 Cancer should be shortened to some degree.
Response: We have removed some portions as well as shortened. Some extra relevant matters were incorporated.
Comment 4: The resolution of figure 7 is not enough.
Response: Figure 7 was edited and now the picture is clear and legible (Page 15).
Comment 5: The author should draw the structure of all the polysaccharides using the same style, but not showing in the review with different styles.
Response: We have corrected all the structures of all the polysaccharides using the same style.
Comment 6: Line 184-186, one recent review (Advanced Materials 33 (18), 2005513) should be included to support such a claim.
Response: This article is included in our review article.
Comment 7: Moderate editing of English language
Response: We sincerely apologize for the inadvertent errors. We have extensively edited our manuscript to limit typographic and grammatical errors.
Reviewer 3 Report
1. Abstract: “Effective cancer treatment is a worldwide problem these days, and subsequent advancements in nanomedicine are critical. Cancer is one of the most common lethal diseases and the main cause of mortality worldwide. Nanotechnology, which is gaining popularity, enables fast-expanding delivery methods in science for curing diseases in a site-specific approach utilizing natural bioactive substances.” The first sentence does not flow well. The problem of effective cancer treatment is a long on-going issue. Why are advancements in nanomedicine critical? The last sentence is inaccurate and a need to expand on natural bioactive substances.
2. Introduction: “The present article provides an overview of the latest research and developments in the field of natural polymeric nano biocomposites, with a particular emphasis on their applications in the controlled and targeted delivery of anticancer drugs.” How are natural polymer nanocomposites sourced and synthesized? Cellulose Based Nanocomposites? Chitin Based Nanocomposites? Applications of Nanocomposites? How are these natural polymer nanocomposites applied to treat cancer?
3. Section on cancer: This is poorly written and requires a more in-depth description of the main issues confronting cancer treatments. What are the challenges? For example, nanopolymers can be selectively designed to alter the pharmacokinetic profile and tissue distribution characteristics of drug delivery vehicles. The size, shape, and surface modifications, all of which alter the pharmacokinetics and intracellular mechanisms, can be chemically modified such that they can have a significant therapeutic impact in vivo. Investigations into the toxic effects following nanopolymer internalization are minimal as many nanopolymers are designed to be inert delivery vehicles with little or no toxic effects when they release their contents. To this end, Albanese et al. provided a detailed review on the effect of the nanoparticle size, shape and surface chemistry on biological systems upon internalization [Annu. Rev. Biomed. Eng. 2012, 14, 1–16, doi:10.1146/annurev-bioeng-071811-150124]. Physical attributes of nanopolymers continued to be explored regarding their effects on therapeutic efficacy; however, the consensus remains that the effects and final properties of nanoparticles in the endo-lysosomal vesicles of cells remain unknown. For example, nanoparticles in the intracellular cytosol space can activate several biological functions, including disrupting mitochondrial function, eliciting the production of reactive oxygen species and activation of the oxidative stress mediated signaling cascade. Other reports have demonstrated that nanoparticles such as hydrophilic titanium oxide TiO2 nanoparticles are oncogenic. It is known that large nanoparticles do not extravasate far beyond the blood vessel, whereas small nanoparticles travel deep into the tumor, but remain there only transiently. Therefore, it is essential to optimize the next generation of nanopolymers focusing on the intracellular therapeutic mechanisms after internalization to successfully translate these drug delivery systems to the clinic.
4. Designing drug delivery vehicles that actively target cancer cells is an area of research interest as an alternative treatment of cancer, providing a better quality of life for cancer patients. However, it is unknown how these targeted delivery vehicles interact with a three-dimensional (3D) tumour mass in vivo. Although numerous delivery vehicles can selectively target cancer cells, the majority of these studies use two-dimensional cellular monolayer systems. These study models do not accurately mimic the complex in vivo interactions that take place between cancer cells and their tumour microenvironment. Indeed, many delivery systems show promise in vitro, but these approaches do not translate similarly to in vivo applications. The ability of delivery drug vehicles to interact with a 3D tumour mass is initially tested in vivo and to monitor the efficacy of this interaction process in real-time is lengthy, expensive, complicated, and challenging. Recent reports have investigated the application of nanoparticles to 3D tumour spheroids to characterize their ability to penetrate the mass of cancer cells [3,4]. However, these studies used nanoparticles that do not actively target overexpressed receptors on cancer cells. As such, investigating the mechanism(s) of action of targeted delivery vehicles and their interaction with 3D multicellular structures serves as an essential intermediate step between in vitro and in vivo studies. To this end, Minchinton and Tannock [Nat. Rev. Cancer 2006, 6, 583–592] reported an eloquent review of the strategies to improve drug penetration through tumour mass and the design of selective compounds that have the targeted abilities to penetrate tissue. It is noteworthy that 100 nm particles or larger generally do not penetrate well throughout the tumour mass, and smaller nanoparticles do not accumulate sufficiently in the tumour vasculature by the enhanced permeability and retention (EPR) effect and do not achieve good tumour penetration.
5. Definition and classification of nano-bio composites. The review manuscript should mainly focus on this section and explore the different applications to treating cancer as outlined in the above comments. Also, see line 1038, “The efficacy of chitosan-starch nanocomposite particles loaded with bis-demethoxy curcumin analog in targeting cancer cells was verified through MTT assays conducted on MCF-7 breast cancer cell lines and VERO cell lines [173].” This sentence does not describe the results whether it worked or not. Unfortunately, the manuscript is written this way and thus requires more in-depth details of the experimental design and the findings. What are the challenges?
6. There are several articles describing the effectiveness of NPs in targeting oncoproteins and penetration of organoids and tumor mass.
7. The manuscript reveals over 1000 grammar errors with 2% plagiarism.
The manuscript reveals over 1000 grammar errors with 2% plagiarism.
Author Response
3rd Reviewer’s Comments:
Comment1: Abstract: “Effective cancer treatment is a worldwide problem these days, and subsequent advancements in nanomedicine are critical. Cancer is one of the most common lethal diseases and the main cause of mortality worldwide. Nanotechnology, which is gaining popularity, enables fast-expanding delivery methods in science for curing diseases in a sitespecific approach utilizing natural bioactive substances.”
The first sentence does not flow well. The problem of effective cancer treatment is a long on-going issue. Why are advancements in nanomedicine critical? The last sentence is inaccurate and a need to expand on natural bioactive substances.
Response: We have made necessary changes in the abstract as per the suggestion of the reviewer.
Comment 2: Introduction: “The present article provides an overview of the latest research and developments in the field of natural polymeric nanobiocomposites, with a particular emphasis on their applications in the controlled and targeted delivery of anticancer drugs.” How are natural polymer nanocomposites sourced and synthesized? Cellulose Based Nanocomposites? Chitin Based Nanocomposites? Applications of Nanocomposites? How are these natural polymer nanocomposites applied to treat cancer?
Response: We have already discussed this part in the section 5, which is related to the natural polymer nanocomposites and their applications. About the chitin and cellulose based nanocomposites, it was mentioned in the section 5.1 and 5.3.
Comment 3: Section on cancer: This is poorly written and requires a more in-depth description of the main issues confronting cancer treatments. What are the challenges? For example, nanopolymers can be selectively designed to alter the pharmacokinetic profile and tissue distribution characteristics of drug delivery vehicles. The size, shape, and surface modifications, all of which alter the pharmacokinetics and intracellular mechanisms, can be chemically modified such that they can have a significant therapeutic impact in vivo. Investigations into the toxic effects following nanopolymer internalization are minimal as many nanopolymers are designed to be inert delivery vehicles with little or no toxic effects when they release their contents. To this end, Albanese et al. provided a detailed review on the effect of the nanoparticle size, shape and surface chemistry on biological systems upon internalization [Annu. Rev. Biomed. Eng. 2012, 14, 1–16, doi:10.1146/annurev-bioeng-071811-150124]. Physical attributes of nanopolymers continued to be explored regarding their effects on therapeutic efficacy; however, the consensus remains that the effects and final properties of nanoparticles in the endo-lysosomal vesicles of cells remain unknown. For example, nanoparticles in the intracellular cytosol space can activate several biological functions, including disrupting mitochondrial function, eliciting the production of reactive oxygen species and activation of the oxidative stress mediated signaling cascade. Other reports have demonstrated that nanoparticles such as hydrophilic titanium oxide TiO2 nanoparticles are oncogenic. It is known that large nanoparticles do not extravasate far beyond the blood vessel, whereas small nanoparticles travel deep into the tumor, but remain there only transiently. Therefore, it is essential to optimize the next generation of nanopolymers focusing on the intracellular therapeutic mechanisms after internalization to successfully translate these drug delivery systems to the clinic.
Response: As instructed by the reviewer, some portions have been removed and it was shortened. Some extra relevant matters were incorporated.
Comment 4: Designing drug delivery vehicles that actively target cancer cells is an area of research interest as an alternative treatment of cancer, providing a better quality of life for cancer patients. However, it is unknown how these targeted delivery vehicles interact with a three-dimensional (3D) tumour mass in vivo. Although numerous delivery vehicles can selectively target cancer cells, the majority of these studies use two-dimensional cellular monolayer systems. These study models do not accurately mimic the complex in vivo interactions that take place between cancer cells and their tumour microenvironment. Indeed, many delivery systems show promise in vitro, but these approaches do not translate similarly to in vivo applications. The ability of delivery drug vehicles to interact with a 3D tumour mass is initially tested in vivo and to monitor the efficacy of this interaction process in real-time is lengthy, expensive, complicated, and challenging. Recent reports have investigated the application of nanoparticles to 3D tumour spheroids to characterize their ability to penetrate the mass of cancer cells [3,4]. However, these studies used nanoparticles that do not actively target overexpressed receptors on cancer cells. As such, investigating the mechanism(s) of action of targeted delivery vehicles and their interaction with 3D multicellular structures serves as an essential intermediate step between in vitro and in vivo studies.
To this end, Minchinton and Tannock [Nat. Rev. Cancer 2006, 6, 583–592] reported an eloquent review of the strategies to improve drug penetration through tumour mass and the design of selective compounds that have the targeted abilities to penetrate tissue. It is noteworthy that 100 nm particles or larger generally do not penetrate well throughout the tumour mass, and smaller nanoparticles do not accumulate sufficiently in the tumour vasculature by the enhanced permeability and retention (EPR) effect and do not achieve good tumour penetration.
Response: Suitable drug delivery therapeutics utilizing natural nano biocomposites were the main focus of our article. Investigating the mechanism(s) of action of targeted delivery vehicles and their interaction with 3D multicellular structures are not related to this article. However, Minchinton and Tannock reference was included in our review article.
Comment 5: Definition and classification of nano-bio composites. The review manuscript should mainly focus on this section and explore the different applications to treating cancer as outlined in the above comments. Also, see line 1038, “The efficacy of chitosan-starch nanocomposite particles loaded with bis-demethoxycurcumin analog in targeting cancer cells was verified through MTT assays conducted on MCF-7 breast cancer cell lines and VERO cell lines [173].” This sentence does not describe the results whether it worked or not. Unfortunately, the manuscript is written this way and thus requires more in-depth details of the experimental design and the findings. What are the challenges?
Response: Rectified and added the results part elaborately.
Comment 6: There are several articles describing the effectiveness of NPs in targeting oncoproteins and penetration of organoids and tumor mass.
Response: Rectified and added some more relevant earlier reported matters and some parts were neglected.
Comment 7: The manuscript reveals over 1000 grammar errors with 2% plagiarism.
Response: We sincerely apologize for the inadvertent errors. We have extensively edited our manuscript to limit typographic and grammatical errors.
Round 2
Reviewer 2 Report
accept based on the revision
Minor editing of English language required
Reviewer 3 Report
perhaps remove or modify fig 2.
OK